# Kalium channelrhodopsins effectively inhibit neurons

Stanislav Ott [1], Sangyu Xu [2], Nicole Lee [1], Ivan Hong [1], Jonathan Anns [2,3], Danesha Devini Suresh [1], Zhiyi Zhang [1], Xianyuan Zhang [1], Raihanah Harion [4], Weiying Ye [5], Vaishnavi Chandramouli [4], Suresh Jesuthasan [4], Yasunori Saheki [4] & Adam Claridge-Chang [1,2,5] ✉

The analysis of neural circuits has been revolutionized by optogenetic methods. Light-gated chloride-conducting anion channelrhodopsins (ACRs)—recently emerged as powerful neuron inhibitors. For cells or sub-neuronal compartments with high intracellular chloride concentrations, however, a chloride conductance can have instead an activating effect. The recently discovered light-gated, potassium-conducting, kalium channelrhodopsins (KCRs) might serve as an alternative in these situations, with potentially broad application. As yet, KCRs have not been shown to confer potent inhibitory effects in small genetically tractable animals. Here, we evaluated the utility of KCRs to suppress behavior and inhibit neural activity in *Drosophila*, *Caenorhabditis elegans*, and zebrafish. In direct comparisons with ACR1, a KCR1 variant with enhanced plasma-membrane trafficking displayed comparable potency, but with improved properties that include reduced toxicity and superior efficacy in putative high-chloride cells. This comparative analysis of behavioral inhibition between chloride- and potassium-selective silencing tools establishes KCRs as next-generation optogenetic inhibitors for in vivo circuit analysis in behaving animals.

The ability to manipulate distinct neuronal populations in a spatiotemporally precise manner is invaluable to research into brain function. A key approach that has revolutionized such research is optogenetics, which uses cell type-specific expression with light gating to precisely control neuronal activity[1–3]. While optogenetic activators have already achieved a high level of potency and sophistication[4–7], their inhibitory counterparts are comparatively less well-developed. Despite progressive improvements[8–10], light-driven inhibitory chloride pumps require high expression levels and strong light intensities[11]. As such, the discovery of a pair of natural chloride-conducting light-gated ion channels[12] represented a major development in inhibitory optogenetics. Isolated from the cryptophyte algae *Guillardia theta*, these

anion channelrhodopsins (ACRs) have proven to be potent and versatile inhibitors of neuronal activity in *Drosophila*[13,14], zebrafish[15], mouse[16–18], ferret[19] and *Caenorhabditis elegans*[20,21].

As an anion channel, the light actuation of an ACR is roughly equivalent to opening a chloride conductance[12] (Fig. 1A). Because the equilibrium potential of chloride in neurons usually falls below the threshold for action potentials, ACR actuation will typically inhibit firing by hyperpolarizing the cell[12,17]. The complexities of chloride physiology, however, mean that chloride-based silencing has at least three relevant caveats. First, the active chloride extrusion found in mature neurons is unusual for animal cells, including both non-excitable and excitable cells, which generally have a high intracellular chloride

[1]Program in Neuroscience and Behavioral Disorders, Duke-NUS Medical School, Singapore, Singapore. [2]Institute for Molecular and Cell Biology, A*STAR Agency for Science, Technology and Research, Singapore, Singapore. [3]School of Biological Sciences and Institute for Life Sciences, University of Southampton, Southampton, UK. [4]Lee Kong Chian School of Medicine, Nanyang Technological University, Singapore, Singapore. [5]Department of Pharmacy, National University of Singapore, Singapore, Singapore. ✉e-mail: claridge-chang.adam@duke-nus.edu.sg

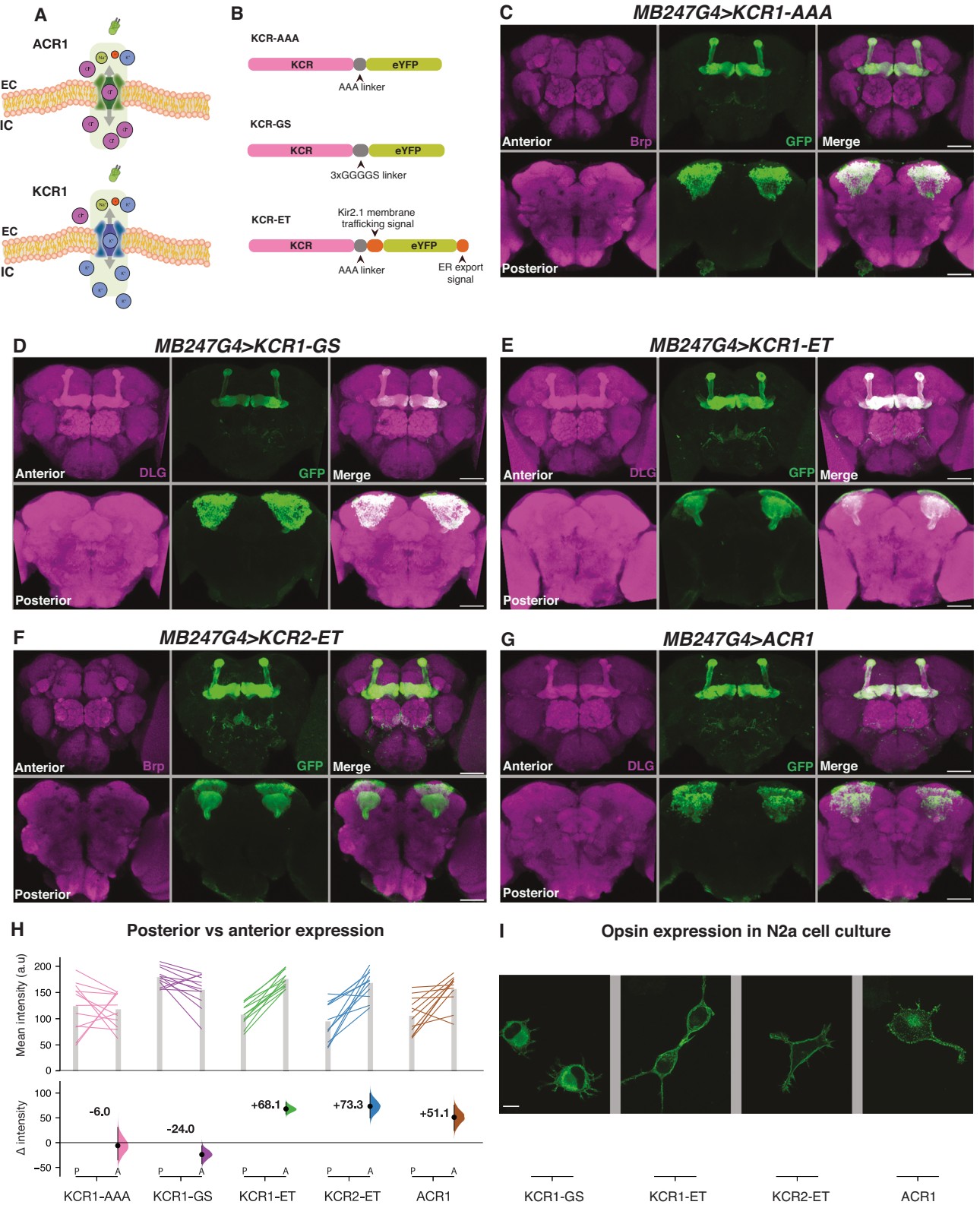

**H** Posterior vs anterior expression

**I** Opsin expression in N2a cell culture

concentration[22-25]. In such cells, a chloride conductance would have a depolarizing effect[26-29] and thus, as has been shown in some cases, rather than inhibiting, ACR actuation can cause activation[11,29,30]. Second, some neuronal compartments (notably axons) have higher steady-state intracellular chloride levels than the soma[31-33]; again here, in such compartments, chloride conductances can be activating[5,11,17,18,34-36]. Third, prolonged chloride conductances can lead to complex secondary effects (including the redistribution of potassium) with diverse impacts on

excitability[22,37]. Careful opsin engineering has reduced ACR activation of axons by targeting opsin expression to the soma[17,18]; however, even this innovation does not resolve the potentially ambiguous effects of optogenetic chloride channels on membrane potential in other contexts.

Potassium (K⁺) channels have fundamental roles in setting the resting membrane potential and terminating action potentials[38]. As such, researchers have long sought a light-actuated K⁺-selective channel to use as a neuronal inhibitor. To this end, chimeric potassium

**Fig. 1 | Membrane-trafficking signals improve KCR localization to axons.**
**A** Schematic of ACR and KCR channelrhodopsins. ACRs are chloride-selective and inhibit spiking via chloride influx. KCRs are potassium-selective and inhibit neuronal activity via the endogenous repolarization process. **B** A schematic of three KCR fusion arrangements; the ET variants contain membrane targeting sequences. **C**–**G** Representative confocal images of fly brains expressing *KCR1-AAA* (**C**), *KCR1-GS* (**D**), *KCR1-ET* (**E**), *KCR2-ET* (**F**), and *ACR1* (**G**) in the mushroom bodies (MB) with *MB247-Gal4*. *ACR1* and the *KCR-ET* variants show robust signals in the axonal MB lobe region, whereas *KCR1-GS* and *KCR1-AAA* show strong somatic signals. Anti-disks large (DLG) or anti-Bruchpilot (Brp) stains are shown in magenta and anti-GFP staining is shown in green. Scale bar = 50 μm. For each genotype, *n* = 1 biologically independent sample over 1 independent experiment. **H** Quantifications of anti-GFP intensity in posterior (P) and anterior (A) brain regions for each opsin transgene crossed with *MB247-Gal4*. Top: Individual brain hemispheres are shown as slope plots. The height of the bars shows average intensity values. Bottom: Posterior–anterior mean differences of anti-GFP intensities; error bars represent the 95% CI. For each genotype, *n* = 12 biologically independent samples over 12 independent experiments. **I** Representative images of opsin expression in transfected N2a cell culture. KCR1-GS displayed stronger cell-interior localization whereas KCR1-ET, KCR2-ET, and ACR1 showed increased plasma-membrane localization. Scale bar = 10 μm. For each genotype, *n* = 1 biologically independent sample over 1 independent experiment. Additional statistical information is presented in Supplementary Dataset 1. Source data are provided as a Source Data file.

channels, such as HyLighter[39], BLINK1[40], BLINK2[41], and PAC-K[42], were engineered to be light-responsive. To date, engineered light-actuated K⁺ channels have relatively slow kinetics, and have not been used in the major invertebrate model systems. Recently, the discovery of genomic sequences from a stramenopile protist led to the identification of two channelrhodopsins that naturally conduct potassium: *Hypochytrium catenoides* kalium channelrhodopsin 1 and 2 (HcKCR1 and HcKCR2, Fig. 1A)[43]. Actuation of these HcKCRs opens K⁺-selective conductances that, as shown for HcKCR1, can inhibit action potentials in mouse brain slices[43]. More recently, HcKCR1 has also been used to successfully suppress neuronal activity during virtual-reality behavior in mice[44]. A third KCR was subsequently identified from the stramenopile *Wobblia lunata*—termed the *Wobblia* inhibitory channelrhodopsin (WiChR)—which has even sharper K⁺ selectivity and inhibits action potentials in anesthetized mouse brain and cardiomyocytes[45].

While KCR efficacy has been shown in brain slices and behaving mice, there are large differences between rodent and invertebrate experiments, including transgene expression levels, membrane targeting, and optical accessibility. As such, we aimed to investigate the utility and efficacy of KCRs to inhibit and silence neurons in vivo in *Drosophila, C. elegans,* and *Danio rerio*—the major small animal models.

## Results

### Trafficking signals improve KCR localization to axons

Heterologous opsins can have poor trafficking to the plasma membrane and are retained in internal membranes[10]. As such, we aimed to identify HcKCR configurations that efficiently localize to neurites. With standard fly transgenic methods, we targeted different KCR fusion proteins (Fig. 1B) to the mushroom bodies (MB)[46,47] and then compared their localization. As the MB somata are located in the posterior brain but their axons project to the anterior MB lobes[47–49], we could use this spatial separation to estimate relative axonal localization. We found that the simplest KCR fusion protein, KCR1 with an enhanced yellow fluorescent protein (eYFP) linked with three alanine residues (KCR1-AAA, Fig. 1B), was equally localized to the MB soma and axons (Fig. 1C, H). This finding is consistent with prior reports showing that HcKCRs have imperfect membrane localization[43,45]. Replacing AAA with a longer linker (3× GGGGS, KCR-GS, Fig. 1B) slightly worsened anterior/axonal localization (Fig. 1D, H). Adding endoplasmic reticulum export and Golgi trafficking (ET) motifs (Fig. 1B)[10,50–53], produced KCR-ET variants with superior relative axonal localization (KCR1-ET and KCR2-ET, Fig. 1E, F, and H). Comparing the GFP signal intensity between anterior and posterior brain regions revealed that KCR1-ET, KCR2-ET, and ACR1 (Fig. 1G), were preferentially localized to axons (Fig. 1H). Expressing these opsin variants in cultured mouse neuroblastoma (N2a) cells[54] confirmed this improvement, revealing a predominantly intracellular localization of KCR1-GS and increased relative membrane localization of KCR1-ET and KCR2-ET (Fig. 1I).

### KCR1 actuation effectively impairs locomotor behavior

Having established the KCR-fusion expression patterns, we next targeted three of the KCR variants and, as a benchmark, ACR1 in

*Drosophila* neurons and tested climbing ability during actuation (Fig. 2A). We used two different drivers: *OK371-Gal4* driving expression in motor neurons and *elav-Gal4* driving expression in all neurons[55,56]. Light actuation had large effects on climbing in all test lines. In *OK371* flies, the strongest effectors were KCR1-ET and ACR1 (Δheight = −37.9 mm and −37.5 mm, respectively; Fig. 2B–E). KCR2-ET was noticeably weaker than the others (Δheight = −22.6 mm). In *elav-Gal4* flies, ACR1 gave the most profound paralysis, while KCR1-ET and KCR1-GS had similarly robust, if incomplete, effects on climbing (Δheight = −53.7, −38.5, and −43.1 mm, respectively, Fig. 2F–H). The results indicate that the blue-light sensitive KCR2 is a weak inhibitor in flies; this may be due to the poor transmission of blue light through the adult fly cuticle[57]. For this reason, we focused our efforts in *Drosophila* on the green-light sensitive KCR1 fusions going forward. Indeed, we saw that KCR1 is a potent optogenetic inhibitor, with different effects with the two drivers: inferior to ACR1 in *elav-Gal4*, but comparable to ACR1 in *OK371-Gal4* cells. Despite their differences in cell-surface localization, we observed no major difference in climbing impairment between KCR1-ET and KCR1-GS in this assay.

To generalize the climbing effects to a different motor assay, we also tested the KCR1 lines in the *OK371-Gal4* motor neurons in a horizontal walking assay (Fig. 2I, J). In controls, light elicited substantial increases in walking speed when comparing light and dark epochs (Fig. 2J, S3). In all test lines, exposure to light resulted in marked declines in walking speed: the light-elicited locomotion reductions were ranked KCR1-GS > KCR1-ET > ACR1; Δspeed = −0.79, −0.45, and −0.33, respectively. In addition to the larger relative speed reductions compared to ACR1 (which were partly due to faster dark-epoch walking in the KCR flies), both KCR lines exhibited near-complete suppression of locomotion: actuated walking speed = 0.5, 0.2, and 0.2 mm/s for ACR1, KCR1-ET, and KCR1-GS, respectively. Along with impaired walking, actuation of either ACR1 or the KCRs in *OK371* motor neurons also induced limb twitching. In ACR1, this did not occur at 44 μW/mm² (Fig. S3 and Supplementary Video SV4), and in both ACR1 and KCR1 lines, resolved during longer exposure (Fig. S3 and Supplementary Video SV1).

Taken together, these findings show that KCR1 actuation in *OK371* neurons suppresses both climbing and horizontal walking, and thus effectively inhibits *Drosophila* motor neuron function. The KCR1 transgenes have comparable performance to ACR1.

### Gustation-dependent feeding and olfactory memory are inhibited by KCR1-ET

We next examined whether KCR1 can be used to inhibit sensory systems. The gustatory receptor Gr64f is expressed by a small cluster of sweet-sensing neurons (Fig. 3A)[58]; inhibiting Gr64f cells with *Gr64f-Gal4* and the potassium rectifying channel Kir2.1 can reduce feeding[59]. As such, we expressed KCR1-ET and ACR1 in these cells to test their ability to attenuate feeding, as analyzed using an automated assay (Espresso, Fig. 3B)[60,61]. Specifically, 24 h-starved flies were allowed to feed for 2 h and were illuminated for the initial 30 min of each hour (Fig. 3C). While illumination had no effect on consumption in controls, both *ACR1* and *KCR1-ET* flies consumed less food during light-on

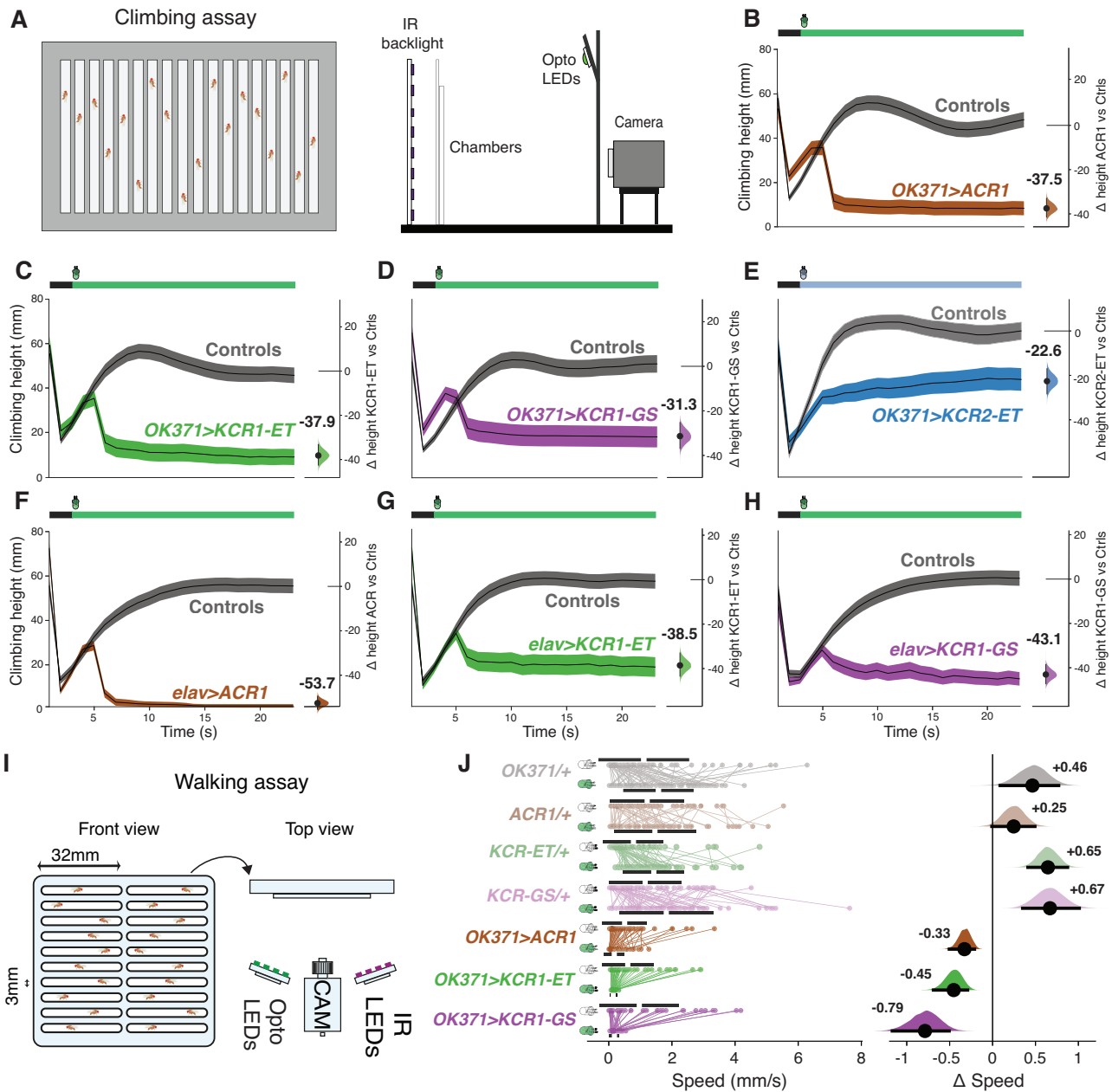

epochs. The feed-volume reductions in ACR1 and KCR1-ET flies were similar (Δvolume = −0.34 μl and −0.30 μl, respectively; Fig. 3D). Thus, KCR1-ET effectively inhibits at least one class of primary-sensory neurons in flies to the same extent as ACR1.

To test KCR1-ET in higher-order sensory neurons, we examined the intrinsic neurons of the MB, which are required for associative olfactory memory[62]. We expressed *KCR1-ET* or *ACR1* in the MB with *MB247-Gal4* and subjected the flies to an aversive Pavlovian conditioning paradigm (Fig. 3E)[13]. During actuation, memory was strongly impaired in both ACR1 and KCR1-ET lines, with conditioned odor preference reduced to near-indifference (ΔPI = −0.31, −0.38, respectively, Fig. 3F, G). During a retest without light, conditioned avoidance was intact thereby attributing the defective recall to channelrhodopsin actuation. These results confirm that ACR1 and KCR1-ET have comparable efficacy for inhibiting sensory neurons.

**Spontaneous action potentials are strongly inhibited by KCR1**
Thus far, we have seen that KCR1 and ACR1 have similar inhibitory effects, except for the *elav-Gal4* climbing experiment, where

ACR1 outperformed KCR1. To investigate this performance difference further, we performed electrophysiological recordings from abdominal nerves in fly larvae with *elav-Gal4* driving KCR1-GS, KCR1-ET, or ACR1. Green light actuation for 30 s produced a strong suppression of spiking for all three genotypes (Fig. 4A, B). These data reveal that all three opsins allowed some residual spiking in some nerves; in all ACR1 recordings there was a rapid (-1 s) and nearly complete inhibition, while in some of the KCR1 recordings, there was a 10–15 s lag before complete inhibition in most nerves (Fig. 4B, C).

In the presence of light, recordings from the *elav/+* controls showed no change; however, in the *ACR1/+* and *KCR1-GS/+* controls there were notable firing-rate dips (Fig. 4D). When we quantified the change in spiking preceding and during illumination, we saw that in flies with *elav-Gal4* driving ACR1, KCR1-GS or KCR1-ET, spikes were suppressed by −100%, −96%, and −93% during the 30 s actuation, respectively (Fig. 4E, F). For the controls, both *elav/+* and *KCR1-ET/+* spike rates were largely unaffected by light: −3% and −3% in the 30 s epoch (Fig. 4E, F). Notably, the *ACR1/+* and *KCR1-GS/+* controls

**Fig. 2 | KCR actuation inhibits climbing and walking in *Drosophila*. A** Schematic of the single-fly climbing assay, showing the chamber on the left and the different assay elements on the right. **B–E** Averaged climbing performance of flies expressing the respective opsin with *OK371-Gal4* and the corresponding Gal4 driver and UAS responder controls (gray) in the presence of light. In the schematic bar (top), black indicates the 3s baseline, and the colored bar indicates the illumination interval. The *y*-axis on the right indicates the mean difference in effect size between the genotypic controls and test flies. The last 10s of the experiment were used for effect size comparisons. Error bands represent the 95% CI. *OK371 > ACR1* test, *n* = 136 biologically independent animals over 8 independent experiments, *OK371 > ACR1* controls, *n* = 255 biologically independent animals over 15 independent experiments. *OK371 > KCR1-ET* test, *n* = 119 biologically independent animals over 8 independent experiments, *OK371 > KCR1-ET* controls, *n* = 238 biologically independent animals over 14 independent experiments. *OK371 > KCR1-GS* test, *n* = 136 biologically independent animals over 8 independent experiments, *OK371 > KCR1-GS* controls, *n* = 221 biologically independent animals over 13 independent experiments. *OK371 > KCR2-ET* test, *n* = 116 biologically independent animals over 7 independent experiments, *OK371 > ACR1* controls, *n* = 136 biologically independent animals over 8 independent experiments. **F–H** Averaged climbing performance of flies expressing the respective opsin with *elav-Gal4* and the corresponding genotypic controls in the presence of light. Green illumination intensity was 11 μW/mm². The blue illumination intensity of KCR2-ET was 85 μW/mm². Error bands represent the 95% CI. *elav > ACR1* test, *n* = 153 biologically independent animals over 9 independent experiments, *elav > ACR1* controls, *n* = 255 biologically independent animals over 15 independent experiments. *elav > KCR1-ET* test, *n* = 135 biologically independent animals over 8 independent experiments, *elav > KCR1-ET* controls, *n* = 255 biologically independent animals over 15 independent experiments. *elav > KCR1-GS* test, *n* = 117 biologically independent animals over 7 independent experiments, *elav > KCR1-GS* controls, *n* = 255 biologically independent animals over 15 independent experiments. **I** Schematic representation of the walking assay with the chamber view from the front (left) and the experimental setup view from the top (right). **J** Walking-speed comparisons before and during illumination (24 μW/mm²). The left side of the plot displays the activity of individual flies (dots), and the gap between the horizontal error bars represents the mean. The right side of the plot displays the speed mean difference effect size for each respective genotype. Error bars represent the 95% CI. *OK371/+*, *n* = 75 biologically independent animals over 3 independent experiments. *ACR1/+*, *n* = 61 biologically independent animals over 3 independent experiments. *KCR1-ET/+*, *n* = 63 biologically independent animals over 3 independent experiments. *KCR1-GS/+*, *n* = 73 biologically independent animals over 3 independent experiments. *OK371 > ACR1, n* = 69 biologically independent animals over 3 independent experiments. *OK371 > KCR1-ET, n* = 57 biologically independent animals over 3 independent experiments. *OK371 > KCR1-GS, n* = 48 biologically independent animals over 2 independent experiments. Additional statistical information for all panels is presented in Supplementary Dataset 1. Source data are provided as a Source Data file.

exhibited a marked drop in the light-dependent firing rate: −51%, and −57% in the 30 s epoch (Fig. 4E, F). Similar effects were observed with the 0.5 s illumination. We interpret the Gal4-independent, off-target effects in the ACR1 and KCR1-GS control lines as the result of leaky expression from the transgenes[63–65]. In conclusion, though they appeared slightly less potent than ACR1, the KCR1 transgenes were highly effective at spiking suppression.

### Increased ATR concentration and light intensity improve KCR inhibitory potency

While vertebrates contain ATR, channelrhodopsin experiments in flies require ATR food supplementation. Structural studies have revealed that the ATR–opsin to pore stoichiometry of ACR1 is 1:1, while the KCRs have an ATR–opsin to pore ratio of 3:1[66–69]. We hypothesized that increased ATR levels might further increase KCR1 efficacy.

In prior experiments we used 0.5 mM ATR food; we therefore repeated the *elav-Gal4* larval nerve recordings using 1 and 2 mM ATR. Because ACR1 actuation with 0.5 mM ATR already completely silenced firing, increasing the ATR concentration had no additional effect on this line (Fig. S2A, B). By contrast, increasing the ATR concentration did improve the potency of KCR1-GS during short actuation periods: in the 0.5 s recordings, suppression improved from 92% to 98% in the 0.5 and 2 mM ATR preparations, respectively (Fig. S2C). An ATR trend was not as pronounced in the longer 30 s actuation recordings. Here, inhibition remained at -97% in the 0.5 and 2 mM ATR preparations, respectively (Fig. S2D–F), presumably because the late-phase suppression (the latter 20 s) was already strong in the lower ATR experiments.

We then examined the effects of changing ATR concentration and light intensity at the behavioral level. In the horizontal walking assay, *OK371-Gal4* expression experiments with increased ATR showed no consistent additional suppression, possibly because ATR is not limiting (Fig. S3 and Supplementary Videos SV1–SV3). However, increasing the light intensity from 24 to 44 μW/mm² rendered all ACR1- and KCR1-expressing flies completely stationary. Any residual speed—more apparent in the KCR1 lines—was an artifact of non-locomotor twitching (Fig. S3 and Supplementary Videos SV5, 6). From these comparisons with ACR1, we conclude that KCR1 efficacy benefits from somewhat higher levels of ATR and stronger light intensities[13].

### KCR1 has limited toxicity

We previously found that adult flies expressing ACR1 did not die earlier than controls[13]. To estimate developmental toxicity in the three channelrhodopsin lines, we established *elav-Gal4* crosses, counted eggs, and maintained them in the dark on normal food without additional ATR supplementation until the offspring emerged. Egg-to-offspring ratios revealed that 47% of *elav > ACR1* eggs failed to develop into adults (Fig. S4A). For *elav > KCR1-ET*, 28% of the eggs failed to develop and for *elav > KCR1-GS* just 13% showed developmental lethality (Fig. S4B, C).

We next used *AstA-Gal4* expression to measure channelrhodopsin toxicity to central brain cells after six days of light exposure[70]. Comparing ACR1, KCR1-ET, or GFP controls, we saw no difference in the number of AstA-positive cells or their morphology (Fig. S4D–F), confirming limited toxicity in adult neurons[13]. We also found that, like ACR1, the KCR1 transgenes were 100% effective at preventing wing expansion (Fig. S4G) via actuation of the bursicon neurons for the four days of metamorphosis[13,71]. These results establish that ACR1 and KCR1 transgenes have comparably low levels of adult toxicity and that KCR1 has lower developmental toxicity.

### Pan-neuronal KCR actuation inhibits movement in *C. elegans*

ACRs have previously been shown to inhibit neuronal activity in *C. elegans*[21]. To test whether KCRs are functional in this model organism, we expressed the opsins in neurons using the pan-neuronal *snt-1P* promoter[72]. Consistent with the observations in fly and N2a cells, the addition of ET sequences also improved membrane targeting in *C. elegans* (Fig. 5A). *ACR1, KCR1-ET,* and *KCR2-ET* worms were cultured on different ATR concentrations prior to locomotor assessment (Fig. 5B). All opsin-expressing worm lines showed movement reductions during illumination, (Supplemental Videos SV7–9). Although the effect sizes varied with ATR concentration, the overall efficacy ranking was KCR2-ET > ACR1 > KCR1-ET (Fig. 5C–E). All animals showed rapid recovery after light exposure. We did not observe major differences in post-illumination recovery time between the opsin-expressing genotypes or between worms grown under different ATR concentrations (Fig. 5F–K). Taken together, these data indicate that, in *C. elegans*, KCR2-ET and ACR1 produce comparable inhibitory effects.

### KCR1 in spinal motor neurons inhibits zebrafish larval movements

We have previously shown that ACRs can be used to inhibit locomotor behavior in zebrafish[15]. To test whether a similar effect can also be achieved with KCRs, we crossed transgenic zebrafish carrying *KCR1-ET*,

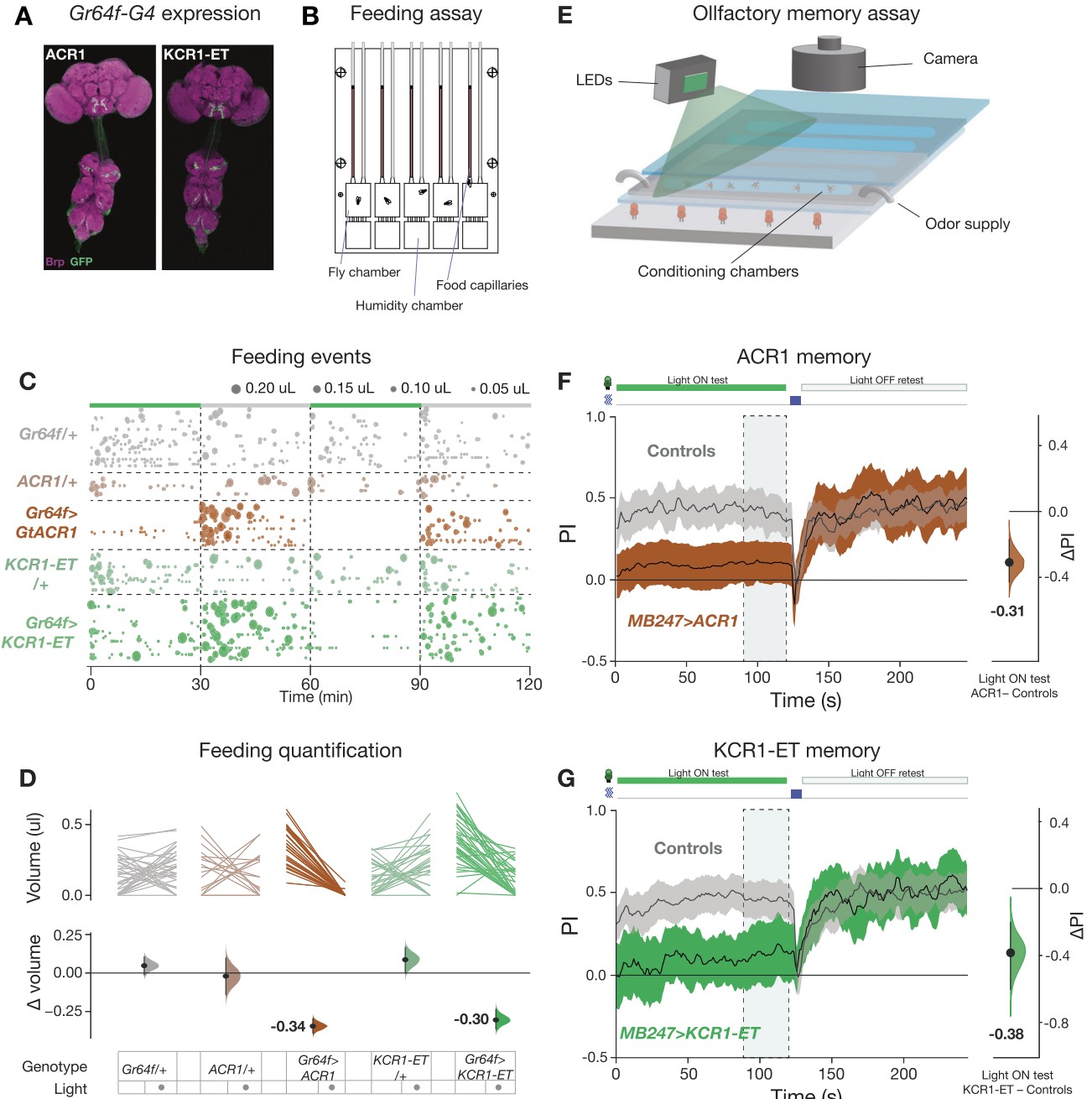

**Fig. 3 | KCR1 and ACR1 actuation show comparable effects on feeding and memory. A** Expression profiles of *Gr64f > ACR1* and *Gr64f > KCR1-ET* flies. Anti-Brp staining is shown in magenta and anti-GFP staining is shown in green. For each genotype, *n* = 1 biologically independent sample over 1 independent experiment. Scale bar = 50 μm. **B** Schematic of the ESPRESSO feeding assay chip. **C** Feeding events in the presence and absence of green light illumination (24 μW/mm²). The schematic bar at the top indicates the illumination epochs in green. The number and size of the bubbles indicate the count and volume of individual feeds, respectively. **D** The top plot displays the averaged paired comparisons of feeding volume between the lights off and on testing epochs. The bottom plot shows the averaged mean difference in feeding volume effect size for the light off and on epochs. Error bars show the 95% CI. *Gr64f/+*, *n* = 36 biologically independent animals over 3 independent experiments. *ACR1/+*, *n* = 18 biologically independent animals over 3 independent experiments. *Gr64f > ACR1*, *n* = 30 biologically independent animals over 3 independent experiments. *KCR1-ET/+*, *n* = 28 biologically independent animals over 3 independent experiments. *Gr64f > KCR1-ET*, *n* = 41 biologically independent animals over 3 independent experiments. **E** Schematic of

the olfactory training assay. **F**, **G** Green light actuation of the MB cells (58 μW/mm²) with *MB247 > ACR1* (**F**) and *MB247 > KCR1-ET* (**G**) strongly impaired shock-odor memory. Retesting the same animals in the absence of illumination restored conditioned shock-odor avoidance. The panels show the dynamic shock-odor avoidance performance index (PI) during the light-on and light-off testing epochs. Flies were agitated by five air puffs between the two testing epochs. The schematic (top) indicates the periods of illumination (green rectangle) and agitation (blue rectangle). The axis on the right shows the mean difference effect size comparison between the PI of the controls and the test genotype. The dashed rectangle indicates the time interval used for effect size comparisons. Error bands represent the 95% CI. *MB247 > ACR1* test, *n* = 228 biologically independent animals over 4 independent experiments, *MB247 > ACR1* controls, *n* = 348 biologically independent animals over 6 independent experiments. *MB247 > KCR1-ET* test, *n* = 216 biologically independent animals over 5 independent experiments, *MB247 > KCR1-ET* controls, *n* = 396 biologically independent animals over 8 independent experiments. Additional statistical information is presented in Supplementary Dataset 1. Source data are provided as a Source Data file.

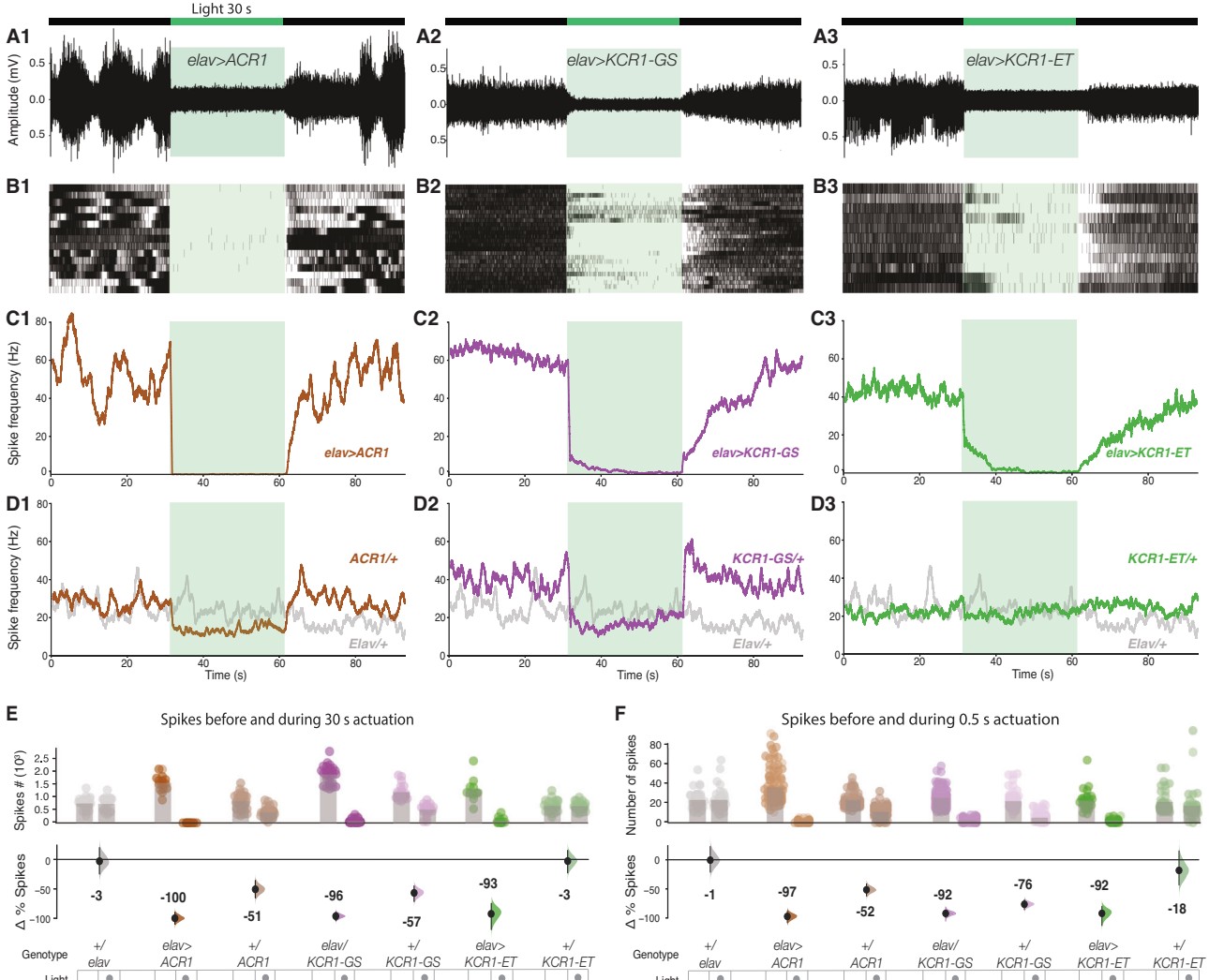

**Fig. 4 | Spontaneous spiking in a larval abdominal nerve is silenced by KCR1.**
**A** Representative traces from extracellular recordings of *Drosophila* larvae abdominal nerve 3. ACR1, KCR1-GS, and KCR1-ET channelrhodopsins were expressed pan-neuronally with *elav-Gal4*. Actuation with green light (40 µW/mm²) was induced for 30 s (schematic on top). For each genotype *n* = 1 biologically independent sample over 1 independent experiment. **B** Rasters of action-potential occurrences in larvae expressing *elav > ACR1*, *elav > KCR1-GS*, and *elav > KCR1-ET*. For each genotype, *n* = 3 biologically independent samples over 3 independent experiments. **C** Averaged spike-frequency charts for the test genotypes. **D** A reduction in action potentials was detected in *ACR1/+* and *KCR1-GS/+* controls, but

not in *KCR1-ET/+* or in *elav/+* controls. For each genotype in **C**, **D**, *n* = 3 biologically independent samples over 3 independent experiments. Error bars show 95% CI. **E** Quantification of spike totals in the 30 s epochs before and during actuation. Each dot in the scatter plot (top) represents the number of spikes per recording. The differences in spike counts before *versus* during actuation are shown (bottom). **F** Quantification of spike totals using 0.5 s light actuation. Error bars in **E** and **F** show 95% CI. For each genotype, *n* = 3 biologically independent samples over 3 independent experiments. Additional information on effect-size statistics is presented in Supplementary Dataset 1. Source data are provided as a Source Data file.

*KCR1-GS*, and *ACR2* with the *s1020t:GAL4* motor neuron driver[15]. The KCR-ET opsin showed expression in motor neurons (Fig. 6A, B). While control animals displayed only mild responses to light, all three opsin-carrying lines showed marked reductions in motor function (Fig. 6C–H). KCR1-GS showed the strongest inhibition of activity, while KCR1-ET and ACR2 showed improved post-actuation recovery. These results suggest that KCR1 and ACR2 have comparable efficacies for inhibiting zebrafish neurons.

**The larval nociceptive threshold is raised by KCR1-ET but not ACR1**

In fly larvae, the multi-dendritic nociceptor neurons mediate nocifensive behaviors[73]. Several classes of these neurons express Subdued, an anoctamin/TMEM16 chloride channel[29,74]. The *subdued* driver *c240-Gal4* targets expression to nociceptor neurons and inhibiting

these cells reduces nocifensive rolling responses to heat[75,76]. In at least one class of Subdued neurons, a chloride current is depolarizing and excitatory[29], suggesting they have high intracellular chloride. To see if Subdued neurons have different responses to KCR actuation, we compared *c240 > ACR1* and *c240 > KCR1-ET* larvae in a temperature ramp. Larvae were placed in a drop of water and gradually heated up in the presence of light. There was no difference in nocifensive response onset between genotypic controls and *c240 > ACR1* larvae. However, the nocifensive onset threshold of *c240 > KCR1-ET* larvae was about +2 °C higher, as compared to the respective genotypic controls (Fig. 7A, B). This finding shows that ACR1 does not affect Subdued-cell physiology, while KCR1-ET can inhibit Subdued nociceptors, which are putative high-chloride cells. This result is consistent with the idea that KCR1 actuation inhibits neurons with a high intracellular chloride concentration.

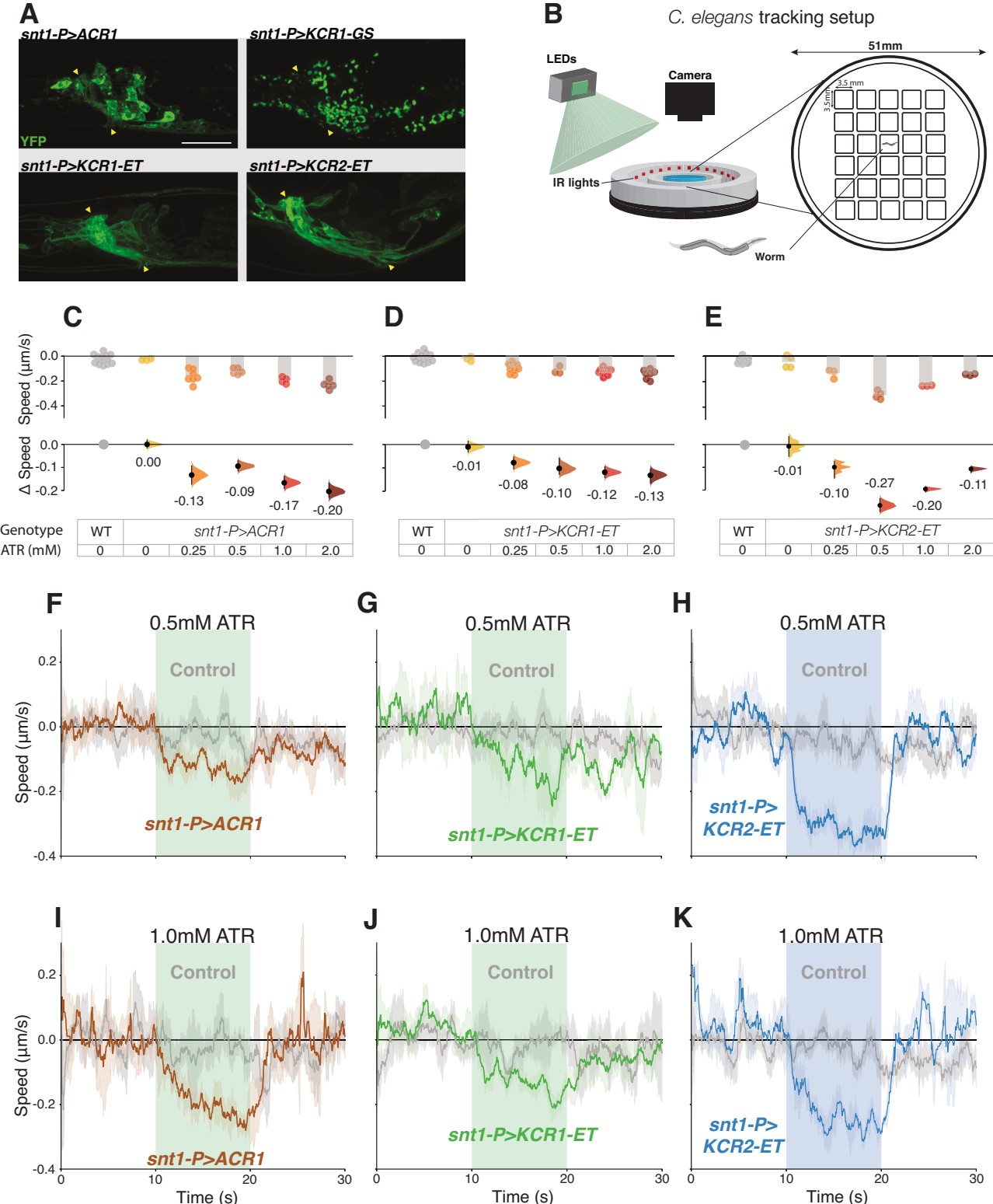

## Actuating Hodor enterocytes with KCR1-ET reduces larval feeding

As chloride conductances can have diverse effects on different cell types, we aimed to investigate the differences between ACR1 and KCR1 actuation in non-neuronal cells that use chloride signaling. We focused on the zinc-gated chloride channel pHCl-2, known as Hodor, which is expressed in a subset of *Drosophila* larval enterocytes[77,78] and lowers cytoplasmic chloride levels[78]. A loss of pHCl-2/Hodor function results in a decrease in feeding, along with a systemic decrease in insulin

signaling[78]. We hypothesized that, as a chloride channel, ACR1 might have an effect similar to a *hodor* gain of function, while a K⁺-selective channel like KCR1 would be expected to be hyperpolarizing[79]. Using *hodor-Gal4*, we expressed ACR1 and KCR1-ET in these cells and subjected larvae to a dye-feeding assay under green light. We noted that *KCR1-GS*/+ and *ACR1*/+ responder controls consumed overall less food than *KCR1-ET*/+ larvae, potentially due to opsin background expression in these genotypes (Fig. 4D). Actuation of ACR1 did not increase feeding, but instead produced an 18% decrease in the amount of dyed

**Fig. 5 | KCR actuation impairs locomotion in *C. elegans*. A** Representative live confocal images of *C. elegans* head regions expressing the respective opsin in neurons; YFP signals are shown in green. Nerve ring regions that contain bundles of neuronal processes are flanked by yellow arrowheads. Scale bar = 20 μm. For each genotype, *n* = 1 biologically independent sample over 1 independent experiment. **B** Schematic representation of the worm tracking chambers and setup. *C. elegans* movement was captured via video acquisition and evaluated by post hoc processing with DeepLabCut (see "Methods"). **C–E** Comparisons of average changes in speed during light actuation (green = 75 μW/mm², blue = 65 μW/mm²) between wild-type controls and opsin-expressing worms at different ATR concentrations. Each dot in the scatter plots (top) represents one worm and the height of the bar shows the average speed. The Δspeed comparisons between wild-type controls and the respective genotype are shown (bottom). The same wild-type controls were shared between the ACR1 and KCR1-ET experiments. Error bars show 95% CI. Wild-type controls for ACR and KCR1-ET *n* = 16 biologically independent animals over 16 independent experiments. *snt1-P > ACR1* 0 mM ATR, *n* = 3 biologically independent animals over 3 independent experiments. *snt1-P > ACR1* 0.25 mM ATR, *n* = 7 biologically independent animals over 7 independent experiments. *snt1-P > ACR1* 0.5 mM ATR, *n* = 6 biologically independent animals over 6 independent experiments. *snt1-P > ACR1* 1 mM ATR, *n* = 4 biologically independent animals over 4 independent experiments. *snt1-P > ACR1* 2 mM ATR, *n* = 5 biologically independent animals over 5 independent experiments. *snt1-P > KCR1-ET* 0 mM ATR, *n* = 3 biologically independent animals over 3 independent experiments. *snt1-P > KCR1-ET* 0.25 mM ATR, *n* = 11 biologically independent animals over 11 independent experiments. *snt1-P > KCR1-ET* 0.5 mM ATR, *n* = 3 biologically independent animals over 3 independent experiments. *snt1-P > KCR1-ET* 1 mM ATR, *n* = 11 biologically independent animals over 11 independent experiments. *snt1-P > KCR1-ET* 2 mM ATR, *n* = 7 biologically independent animals over 7 independent experiments. Wild-type controls for KCR2-ET, *n* = 13 biologically independent animals over 13 independent experiments. *snt1-P > KCR2-ET* 0 mM ATR, *n* = 4 biologically independent animals over 4 independent experiments. *snt1-P > KCR2-ET* 0.25 mM ATR, *n* = 3 biologically independent animals over 3 independent experiments. *snt1-P > KCR2-ET* 0.5 mM ATR, *n* = 4 biologically independent animals over 4 independent experiments. *snt1-P > KCR2-ET* 1 mM ATR, *n* = 3 biologically independent animals over 3 independent experiments. *snt1-P > KCR2-ET* 2 mM ATR, *n* = 3 biologically independent animals over 3 independent experiments. **F–K** The panels show the average speed of *C. elegans* 10 s before, during, and after actuation. The average crawling speed of worms grown with 0.5 mM ATR (**F–H**) or 1 mM ATR (**I–K**) are shown in the top and bottom panels, respectively. Error bands represent the 95% CI. For ACR 0.5 mM ATR control, *n* = 9 biologically independent animals over 9 independent experiments. For *snt1-P > ACR1* 0.5 mM ATR, *n* = 6 biologically independent animals over 6 independent experiments. For KCR1-ET 0.5 mM ATR control, *n* = 5 biologically independent animals over 5 independent experiments. For *snt1-P > KCR1-ET* 0.5 mM ATR, *n* = 4 biologically independent animals over 4 independent experiments. For KCR2-ET 0.5 mM ATR control, *n* = 8 biologically independent animals over 8 independent experiments. For *snt1-P > KCR2-ET* 0.5 mM ATR, *n* = 3 biologically independent animals over 3 independent experiments. For ACR 1 mM ATR control, *n* = 11 biologically independent animals over 11 independent experiments. For *snt1-P > ACR1* 1 mM, *n* = 5 biologically independent animals over 5 independent experiments. For KCR1-ET 1 mM ATR control, *n* = 8 biologically independent animals over 8 independent experiments. For *snt1-P > KCR1-ET* 1 mM ATR, *n* = 4 biologically independent animals over 4 independent experiments. For KCR2-ET 1 mM ATR control, *n* = 9 biologically independent animals over 9 independent experiments. For *snt1-P > KCR2-ET* 1 mM ATR, *n* = 3 biologically independent animals over 3 independent experiments. Additional statistical information for all panels is presented in Supplementary Dataset 1. Source data are provided as a Source Data file.

food ingested (Fig. 7C, D). By contrast, actuation of *hodor > KCR1-ET* (51 μW/mm² green light) elicited a robust 68% decrease in feeding (Fig. 7D). Thus, while ACR1 had a limited impact, KCR1 had an effect consistent with impaired Hodor-enterocyte signaling[78].

## KCRs with improved K⁺ selectivity localize to neuronal plasma membranes

Although KCRs preferentially conduct K⁺ they also show residual Na⁺ conductance, leading to major efforts to develop KCRs with improved potassium selectivity[45,66]. Several alterations in the pore, including a C29D mutation, were shown to substantially improve the HcKCRs' K⁺ selectivity[45]. We introduced the C29D mutation into the KCR1-ET construct and generated transgenic flies. In parallel, we generated flies that express the recently discovered *W. lunata* KCR, named WiChR[45], which has the highest K⁺ selectivity of all KCRs to date. KCR1-C29D and WiChR showed preferential axonal localization when expressed in *Drosophila* MB (Fig. 8A–C). Their axonal localization was comparable to that of KCR-ET variants (Figs. 1H and 8C). In N2a neuroblastoma cells, KCR1-C29D and WiChR were both somewhat localized to the plasma membrane. Intracellular GFP-positive puncta were more abundant in WiChR cells as compared to KCR1-C29D (Fig. 8D).

## KCR1-C29D and WiChR provide stronger inhibition of climbing behavior

We examined the capacity of KCR1-C29D and WiChR to inhibit a variety of *Drosophila* behaviors. When expressed either pan-neuronally with *elav-Gal4*, or in motor neurons with *OK371-G4*, both opsins were very effective at inhibiting climbing (Fig. 8E). Compared to KCR1-GS, KCR1-ET, and ACR1, the inhibition of climbing behavior with KCR1-C29D and WiChR was markedly stronger (compared Fig. 8E with 2B–H). However, the inhibitory effects on odor memory (Fig. 8F, G) and feeding (Fig. 8H and S5A) were largely similar across all opsins, suggesting that, in these neuronal sets, the full inhibitory capacity had already been reached by using ACR and the earlier KCR variants (compared Figs. 8F–H with 3C–G).

## WiChR flies display delayed post-actuation recovery

In the climbing experiment, WiChR flies displayed a post-actuation recovery that was delayed. In contrast to KCR1-C29D, *Drosophila* expressing WiChR also failed to display aversive odor memory during the unactuated, no-light, retest epoch (Fig. 8F, G). To further characterize this slow recovery, we compared locomotor activity in fly larvae and adults expressing the different opsins. Except for KCR1-ET, we did not observe major differences in activity before or during opsin actuation in larvae and adults expressing the opsins pan-neuronally (Fig. S5A, B). Even though *elav > KCR1-ET* adult flies remained immobile during light exposure, they displayed occasional twitching which was recorded as bursts of activity by the tracking system (Fig. S5A). After light exposure, the activity of *elav > WiChR* flies remained low and recovered slower than in flies expressing the other opsins (Fig. S5B, C). ACR and HcKCR flies also regained their upright posture immediately after the light was switched off, while most WiChR flies remained lying on their back for a prolonged period of time. *OK371 > WiChR*- flies also displayed a delay in post-inhibition recovery (Fig. 8I). Unlike ACR flies and KCR1-ET, which recovered completely in 2 s and 30 s, respectively, more than 70% of flies expressing WiChR failed to recover their locomotor function in the first 60 s following actuation (Fig. 8 J1, 2). These data indicate that WiChR flies display comparatively slow post-actuation recovery.

## WiChR has a potent effect on NKCC+ cells

In *Drosophila*, the intracellular chloride concentration is largely regulated by the cation chloride cotransporters Na⁺ K⁺ Cl⁻ (NKCC) and K⁺ Cl⁻ (KCC)[80,81]. KCC expression lowers intracellular chloride; NKCC expression is associated with intracellular chloride elevation[82,83]. To test whether K⁺- and Cl⁻-selective opsins would elicit different phenotypes in cells where the NKCC transporter is active, we expressed WiChR, KCR1-C29D, and ACR1 with *NKCC-Gal4*. *NKCC > ACR1* expression was largely toxic, however we were able to recover some adult flies for climbing assays. Interestingly, we observed that *NKCC-G4 > ACR1* actuation did not impair climbing performance (Δheight = +2.6, Fig. 8K1). Exposing *NKCC > KCR1-C29D* flies to light-induced twitching and some falling but the effect was overall mild (Δheight = −6.1, Fig. 8K2).

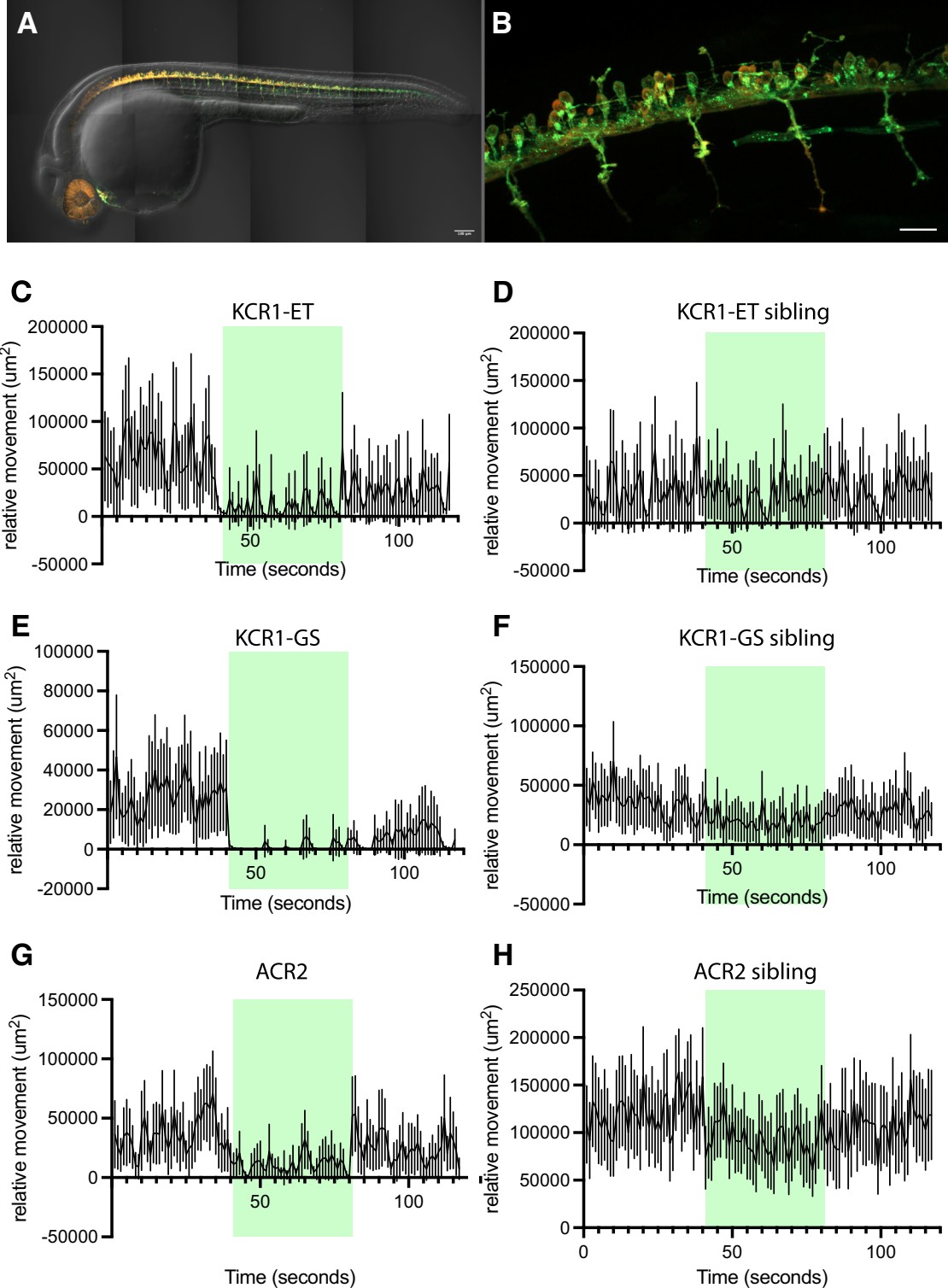

**Fig. 6 | KCR1 actuation inhibits zebrafish larval movements. A** Zebrafish embryos showing expression of the jRGECO1a red fluorescent protein and KCR1-ET with a YFP tag. The proteins were expressed in all spinal motor neurons under the control of the *GAL4s1020t* driver. Scale bar = 100 μm, *n* = 1 biologically independent sample over 1 independent experiment. **B** KCR1-ET is expressed in cell bodies and neurites, with very sparse expression in muscle cells. Scale bar = 100 μm, *n* = 1 biologically independent sample over 1 independent experiment. **C, D** Relative movement of KCR1-ET embryos and non-expressing sibling controls before, during, and after illumination with green light, as indicated by the green

panels. Movement in embryos expressing KCR1-ET is suppressed by green light **E, F** KCR1-GS embryos and control siblings: the former show a pronounced suppression of movement. **G, H** Embryos expressing ACR2 and non-expressing sibling controls. Movement in the ACR2 embryos is suppressed by green light. Line plots show mean relative movement per second with 95% CI error bands. For all genotypes shown in panels (**C**–**H**), *n* = 50 biologically independent animals over 50 independent experiments. Additional statistical information for all panels is presented in Supplementary Dataset 1. Source data are provided as a Source Data file.

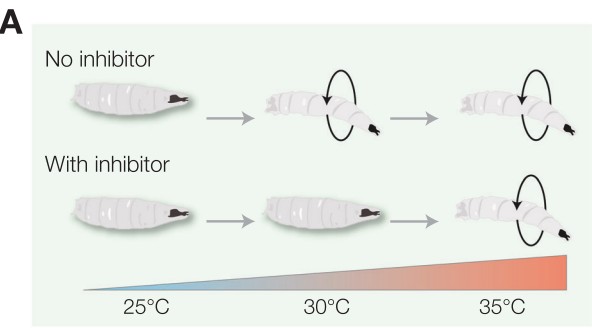

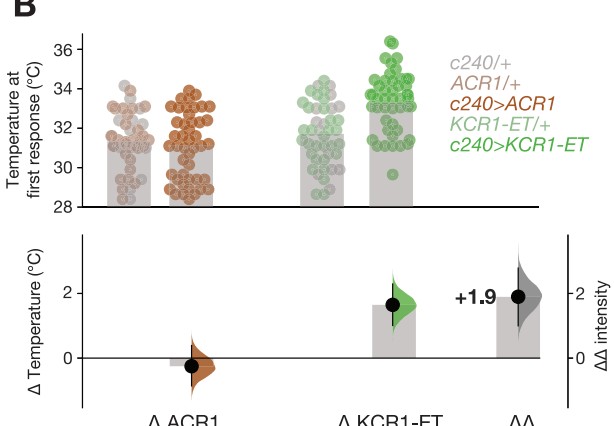

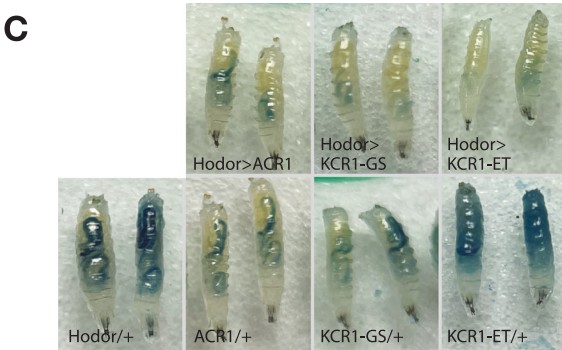

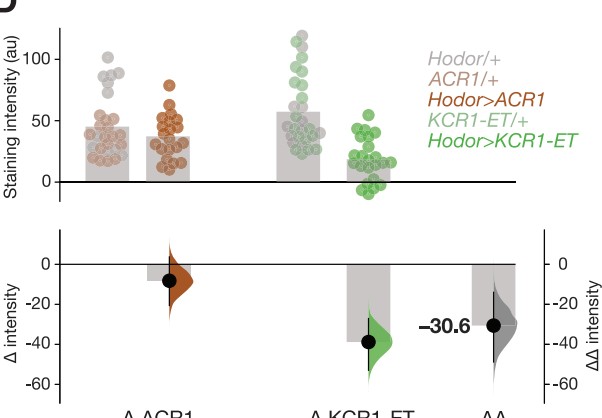

**Fig. 7 | KCR1-ET affects cells with non-canonical chloride signaling. A** Schematic of the heating assay: heating larvae results in a corkscrew nocifensive motor response; inhibiting nociceptors could delay response onset. **B** Onset of the initial nocifensive response of larvae during heating. Actuating *c240 > KCR1-ET* larvae with green light (51 μW/mm²) raised the nocifensive threshold temperature. The plot shows the observed values (top), mean differences, and ΔΔvalues (bottom). Error bars represent the 95% CI. Genotypic controls, *n* = 63 biologically independent animals over 1 independent experiment. *c240 > ACR1*, *n* = 45 biologically independent animals over 1 independent experiment. Genotypic controls *n* = 60 biologically independent animals over 1 independent experiment. *c240 > KCR1-ET*, *n* = 47 biologically independent animals over 1 independent experiment. **C** Representative images of third-instar larvae after ingestion of dyed food. *Hodor/+* = 22 biologically independent animals over 2 independent experiments. *KCR1-ET/+* = 17 biologically independent animals over 2 independent experiments. *KCR1-GS/+* = 9 biologically independent animals over 2 independent experiments. *ACR1/+* = 16 biologically

independent animals over 2 independent experiments. *Hodor > ACR1* = 21 biologically independent animals over 2 independent experiments. *Hodor > KCR1-GS* = 12 biologically independent animals over 2 independent experiments. *Hodor > KCR1-ET* = 23 biologically independent animals over 2 independent experiments. **D** Dye intensity comparisons between controls, *hodor > ACR1,* and *hodor > KCR1-ET* larvae. The top axes show the dye staining intensity. Each dot represents one larva and the bar indicates the mean intensity. The bottom axes show the mean difference effect sizes and relative overall decrease (ΔΔ) between the two opsins. Error bars show the 95% CI. Genotypic controls for ACR, *n* = 27 biologically independent animals over 2 independent experiments. *Hodor > ACR1* = 21 biologically independent animals over 2 independent experiments. Genotypic controls for KCR1-ET, *n* = 23 biologically independent animals over 2 independent experiments. *Hodor > KCR1-ET* = 28 biologically independent animals over 2 independent experiments. Additional statistical information for all panels is presented in Supplementary Dataset 1. Source data are provided as a Source Data file.

By contrast, actuating *NKCC > WiChR* resulted in partial paralysis and a substantial climbing impairment (Δheight = −43.6, Fig. 8K3). The difference between KCR1-C29D and WiChR implies that the latter is more potent, either due to better K+ selectivity and/or higher conductance[45]. Taken together, our analyses of Subdued+, Hodor+, and NKCC+ cells show that ACR-mediated silencing is ineffective in cells with presumptively high intracellular chloride. Our studies have shown that, compared to a chloride channel, potassium-selective inhibitors are comparably potent, less toxic, and more broadly applicable.

## Discussion
Considering the limitations of existing inhibitory optogenetics tools, potassium-conducting channels with rapid light actuation and large currents have been a much sought-after optogenetic tool to reversibly inhibit neural activity. The recent discovery of KCRs, naturally occurring K channels, could represent the next step forward—if KCRs are proven to be effective inhibitors of behavior that match or even surpass the currently existing tools. In the present work, we provide a behavioral and physiological characterization of KCRs in the major small-animal models *Drosophila*, *C. elegans*, and *D. rerio*. Our study had

three primary goals: develop and test KCR transgenic animals, benchmark KCR performance against ACR1 in vivo, and investigate KCR function in high-chloride cells.

## Comparing KCR1 and KCR2
In the first tests in adult flies, we saw that the green-peaked KCR1 was more effective than the blue-peaked KCR2[43]. This finding is consistent with direct measurements of fly cuticles showing that green light has 3× higher penetrance than blue light[84], and observations that red- and green-sensitive channelrhodopsins are more effective at lower light intensities than blue-peaked channelrhodopsins in the adult fly[13,57,84]. While KCR1 was more effective in flies, in *C. elegans*, a transparent animal, we observed that the KCR2 transgene was noticeably more effective than KCR1. Further work would be needed to understand this difference.

## Export and trafficking peptides improve plasmalemmal localization, but not efficacy
Prior work has shown that adding endoplasmic reticulum export and plasma-membrane trafficking (ET) signals to opsins can dramatically (3×) improve surface expression[10]. Comparing simple linkers (AAA and GS) with the ET constructs, we found that the latter improved plasma-

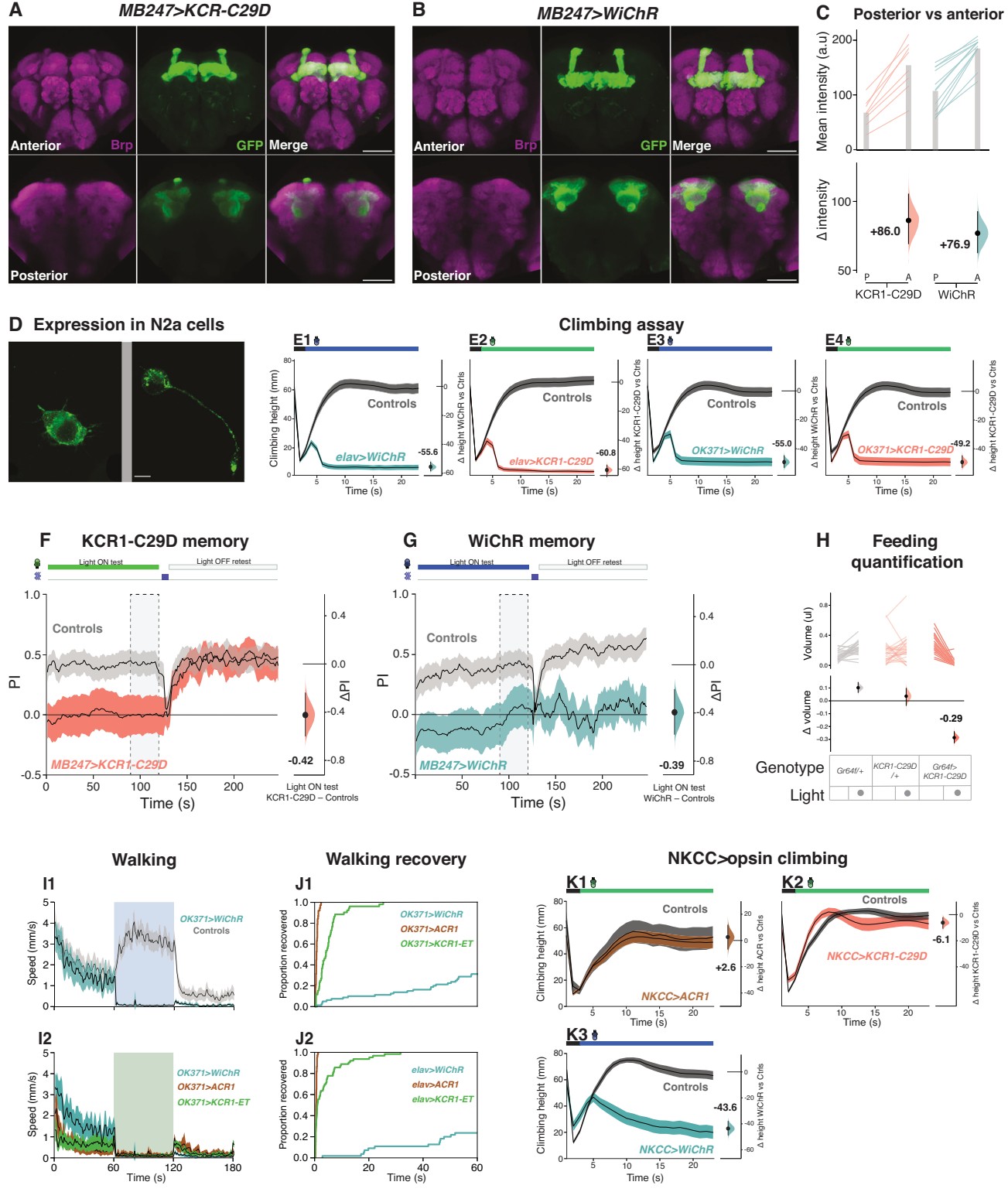

membrane localization of KCRs in fly neurons, mouse neuroblastoma cells, and *C. elegans* neurons (Figs. 1 and 5A). However, despite the marked improvement in surface localization, KCR1-ET performance was not always superior to KCR1-GS, which may be in part due to the off-target expression of the KCR1-GS transgene.

**Benchmarking KCRs against ACR1**

ACR1 is an effective inhibitor in diverse neural systems with presumptively low chloride concentrations[13,14]. Here, we saw that across

a panel of assays in three species, monitoring behaviors and nerve spiking, ACR1 and the KCRs had generally comparable performance.

Specifically, in two experiments using *elav-Gal4*, the climbing assay (Fig. 2) and the nerve recordings (Fig. 3), the suppression of activity or spiking occurred slightly faster and was slightly more consistent with ACR1 than KCR. Increasing the light intensity and ATR concentration yielded improvements in KCR1's silencing speed and completeness, indicating that (compared to ACR1) KCR1 requires

**Fig. 8 | KCRs with improved K⁺ selectivity have increased potency.**
**A, B** Representative confocal images of fly brains carrying **A** *MB247 > KCR1-C29D* or
**B** *MB247 > WiChR*. Both opsins are strongly expressed in the MB lobe axonal
regions. Anti-Brp staining is shown in magenta and anti-GFP staining is shown in
green. Scale bar = 50 µm. For both genotypes, $n = 1$ biologically independent
sample over 1 independent experiment. **C** Quantifications of anti-GFP intensity in
posterior (P) and anterior (A) brain regions for *KCR1-C29D* and *WiChR* crossed with
*MB247-Gal4*. Top: individual brain hemispheres are shown as slope plots. The
height of the gray bars shows average intensity values. Bottom: posterior–anterior
mean difference effect sizes of anti-GFP intensities; error bars represent the 95% CI.
*MB247 > KCR1-C9D*, $n = 4$ biologically independent samples over 4 independent
experiments. *MB247 > WiChR*, $n = 6$ biologically independent samples over 6
independent experiments. **D** Representative images of the opsins in N2a cells.
(Left) *KCR1-C29D* and (Right) *WiChR* showed expression at the membrane, along
with some intracellular puncta. Scale bar = 10 µm. For both genotypes, $n = 1$ bio-
logically independent sample over 1 independent experiment. **E** Climbing perfor-
mance of *WiChR* (**E1** and **E3**) and *KCR1-C29D* (**E2** and **E4**) flies in the presence of
light. The opsins were expressed pan-neuronally (*elav-Gal4*, E1-2) or in motor
neurons (*OK371-Gal4*, E3-4). The performance of opsin-expressing flies was com-
pared with the averaged performance of the corresponding Gal-4 driver and UAS
responder controls (gray) in the presence of light. The last 10 s of the experiment
were used for effect-size comparisons. Error bars represent the 95% CI. Green
illumination intensity was 11 µW/mm². Blue illumination was 85 µW/mm². Geno-
typic controls for *elav > WiChR*, $n = 209$ biologically independent animals over 14
independent experiments. *Elav > WiChR*, $n = 193$ biologically independent animals
over 13 independent experiments. Genotypic controls for *elav > KCR1-C29D*,
$n = 250$ biologically independent animals over 16 independent experiments.
*Elav > KCR1-C29D*, $n = 156$ biologically independent animals over 11 independent
experiments. Genotypic controls for *OK371 > WiChR*, $n = 211$ biologically indepen-
dent animals over 14 independent experiments. *OK371 > WiChR*, $n = 176$ biologi-
cally independent animals over 11 independent experiments. Genotypic controls
for *OK371 > KCR1-C29D*, $n = 278$ biologically independent animals over 18 inde-
pendent experiments. *OK371 > KCR1-C29D*, $n = 176$ biologically independent ani-
mals over 10 independent experiments. **F, G** Inhibiting MB neurons with
*MB247 > KCR1-C29D* (**F**) or *MB247 > WiChR* (**G**) impaired shock-odor memory.
Retesting the same animals in the absence of illumination restored conditioned
odor avoidance (PI) in KCR1-C29D flies. The performance of WiChR-expressing flies
remained low during retest. Green light illumination intensity was 58 µW/mm². Blue
light illumination was 21 µW/mm². Error bands show 95% CI. Genotypic controls for
*MB247 > KCR1-C29D*, $n = 600$ biologically independent animals over 12 indepen-
dent experiments. *MB247 > KCR1-C29D*, $n = 354$ biologically independent animals
over 7 independent experiments. Genotypic controls for *MB247 > WiChR*, $n = 528$
biologically independent animals over 11 independent experiments. *MB247 >
WiChR*, $n = 312$ biologically independent animals over 6 independent experiments.
**H** The top panel displays the averaged paired comparisons of feeding volume
between the lights off and on testing epochs for *Gr64f > KCR1-C29D* flies and

genotypic controls. The bottom panel shows the averaged mean difference in
feeding volume effect size for the light off and on epochs. Green light illumination
intensity was 24 µW/mm². Error bars show 95% CI. *KCR1-C29D/+*, $n = 27$ biologically
independent animals over 3 independent experiments. *Gr64f/+*, $n = 29$ biologically
independent animals over 3 independent experiments. *Gr664f > KCR1-C29D*, $n = 33$
biologically independent animals over 3 independent experiments. **I** The top panel
(**I1**) shows the averaged horizontal walking speed of *OK371 > WiChR* flies and
genotypic controls before, during (indicated by blue and green boxes), and after
light actuation. The bottom panel (**I2**) shows the speed of the same *OK371 > WiChR*
flies and speeds for flies expressing *OK371 > ACR1* and *OK371 > KCR1-ET*. Error
bands show a 95% CI. Green light illumination intensity was 24 µW/mm². Blue light
illumination was 24 µW/mm². Genotypic controls for WiChR, $n = 126$ biologically
independent animals over 7 independent experiments. *OK371 > WiChR*, $n = 80$
biologically independent animals over 5 independent experiments. *OK371 > ACR1*,
$n = 72$ biologically independent animals over 8 independent experiments.
*OK371 > KCR1-ET*, $n = 58$ biologically independent animals over 6 independent
experiments. **J** Kaplan–Meier post-actuation recovery plots for flies expressing
opsins in motor neurons (*OK371-Gal4*, **J1**) or pan-neuronally (*elav-Gal4*, **J2**). All flies
expressing ACR1 recovered in the first 2 s. The majority of KCR1-ET flies recovered
in the first 10 s and the majority of WiChR-expressing flies remained immobile
>60 s after illumination. *OK371 > WiChR*, $n = 80$ biologically independent animals
over 1 independent experiment. *OK371 > ACR1*, $n = 72$ biologically independent
animals over 1 independent experiment. *OK371 > KCR1-ET*, $n = 58$ biologically
independent animals over 1 independent experiment. *Elav > WiChR*, $n = 55$ biolo-
gically independent animals over 1 independent experiment. *Elav > ACR1*, $n = 64$
biologically independent animals over 1 independent experiment. *Elav > KCR1-ET*,
$n = 66$ biologically independent animals over 1 independent experiment.
**K** Climbing performance of *NKCC > ACR1* (**K1**), *NKCC > KCR1-C29D* (**K2**), and
*NKCC > WiChR* (**K3**) flies and their respective genotypic controls (gray) during light
illumination. Exposing *NKCC > ACR1* flies to light did not impair climbing perfor-
mance. Light exposure induced twitching behavior and occasional falls in
*NKCC > KCR1-C29D* flies. Overall, the light effect was not sufficiently strong to
induce substantial climbing impairment. During illumination, *NKCC > WiChR* flies
displayed twitching, falls, and partial paralysis which led to a strong reduction in
climbing. The last 10 s of the experiment were used for effect size comparisons.
Error bands represent the 95% CI. Green light illumination intensity was 11 µW/
mm². Blue light illumination was 85 µW/mm². Genotypic controls for ACR, $n = 201$
biologically independent animals over 12 independent experiments. *NKCC > ACR1*,
$n = 41$ biologically independent animals over 6 independent experiments. Geno-
typic controls for KCR1-C29D, $n = 204$ biologically independent animals over 12
independent experiments. *NKCC > KCR1-C29D*, $n = 102$ biologically independent
animals over 6 independent experiments. Genotypic controls for WiChR, $n = 204$
biologically independent animals over 12 independent experiments. *NKCC >
WiChR*, $n = 97$ biologically independent animals over 6 independent experiments.
Additional statistical information for all panels is presented in Supplementary
Dataset 1. Source data are provided as a Source Data file.

somewhat higher light levels and ATR for maximal inhibition. How-
ever, these *elav-Gal4* experiments were unique in showing the
ACR1 > KCR1 trend; parallel experiments in five other fly systems,
*C. elegans*, and zebrafish showed that the KCRs have comparable effi-
cacy. The differences between *elav-Gal4* and the other experiments
suggest that opsin efficacy might be influenced by the driver and/or
target-cell population, through an unknown mechanism. One possible
cause for apparent ACR1 potency can be at least partly attributed to
the off-target effects seen in nerve recordings from ACR1 (and also
KCR1-GS) controls. In these measurements, the KCR1-ET transgene had
the slowest onset, but it also had the cleanest control effect. These
results reveal that ACR1 efficacy is in part due to leak effects; they also
highlight the relevance of using an all-*trans* retinal-positive (ATR+)
channelrhodopsin control as well as an ATR+ driver control to fully
account for an optogenetic effect. Moreover, minor differences of
speed and efficacy between opsins are not relevant to the great
majority of well-controlled experiments in invertebrate neuroscience,
which typically use epochal (not pulsed) inhibition. We conclude that,
even in canonical, mature, low-chloride neurons, KCRs meet the
benchmark for silencing efficacy.

Another point to consider is post-inhibition recovery. Although
ACR1 actuation in *Drosophila* is associated with a faster recovery
compared to the HcKCRs (Fig. 8J), the overall recovery differences
between ACR1 and KCR1-ET were relatively minor. For many experi-
mental applications, such modest recovery differences will not be
relevant. On the other hand, the recovery delay of WiChR was sub-
stantial (Fig. 8G, J1–2), with a lingering effect on behavior that lasted
minutes. Since endogenous K+ conductances are not typically asso-
ciated with such long-lasting silencing, and WiChR closes within sec-
onds of light-off[45], the cause of such a long-lasting effect remains to be
fully accounted for.

### Off-target inhibition and toxicity effects
In the *elav-Gal4* larval nerve recordings, the driver-less *UAS-ACR1* and
*UAS-KCR1-GS* control flies exhibited substantially inhibited firing upon
illumination (Fig. S1). This control effect was not observed in *UAS-
KCR1-ET* flies. We hypothesize that this off-target conductance arises
from the leaky expression of the channel from these two UAS trans-
genes. We can also infer, therefore, that the *elav-Gal4* experiment
effects are partly attributable to this off-target conductance, and thus

that the apparent *elav > ACR1* potency must be discounted accordingly. Why we see this effect with the *UAS-ACR1* and *UAS-KCR1-GS* transgenes, but not the *UAS-KCR1-ET*, is unclear and requires further investigation going forward. Regardless, this surprising observation indicates that the KCR1-ET transgene is the cleanest intervention and clarifies the need to employ ATR+ *UAS* controls. Moreover, our experiments to evaluate toxicity showed that KCR1 was either comparable or superior to ACR1. Even in the dark, 47% of *elav > ACR1* embryos did not develop, while KCR1 lethality was limited to half that at most (27% and 13% in the KCR1-ET and KCR1-GS lines, Fig. S4).

## Advantages of KCRs

Unlike vertebrate neurons, the activating effects of ACRs have not yet been recorded in fly neurons[13,14]. From our findings, it seems that in most *Drosophila* neurons, an ACR chloride conductance will be hyperpolarizing and/or open an inhibitory shunt current. Nevertheless, evidence from our Subdued-cell (Fig. 7B), Hodor-cell (Fig. 7C, D), and *NKCC-Gal4* (Fig. 8K) experiments showed that, compared to ACRs, using KCRs provides a more versatile, less ambiguous method of optogenetic inhibition. In each case, ACR1 failed to elicit any behavioral effect, while KCR1 actuation produced a phenotype that was consistent with inhibition. Namely, actuation of the Subdued neurons with KCR1 raised the temperature of nocifensive behavior, presumably through silencing of the heat-responsive multi-dendritic cells. Similarly, while ACR1 actuation had only a trivial effect on feeding, the actuation of KCR1 had a phenotype very similar to the loss of the endogenous pHCl-2/Hodor chloride channel. The inhibition-consistent effect in Hodor enterocytes suggests that KCRs will be useful in non-neuronal cells which typically have high intracellular chloride levels[22,23], such as cardiomyocytes and enteroendocrine cells[45,85,86]. Optogenetically actuating *NKCC > ACR1* cells also failed to elicit any behavioral effects, whereas WiChR actuation produced a strong climbing impairment phenotype that is compatible with inhibition (Fig. 8K). This example demonstrates that the actuation of ACR1 in a subset of NKCC+ neurons could result in a potentially misleading false negative. Surprisingly, *NKCC > KCR1-C29D* actuation produced only relatively mild effects compared to WiChR, suggesting that potassium selectivity, channel conductance, and open-state lifetime are relevant for potent silencing of presumptively high-chloride *NKCC+* cells[45].

Together, the ACR1–KCR1 differences demonstrate that, in cases where targeted cells have either non-canonical or unknown chloride physiology, KCRs will be more reliable than ACRs. As the chloride states of cell types are typically not known, ACR actuation has associated ambiguity. For example, if a driver targets a complex population, ACR actuation could potentially drive both activating and inhibiting effects in different cell types. Applying KCRs as first-line inhibitors would avoid such ambiguity.

## Recommendations

Although the newly discovered WiChR KCR possesses superior potency, the long recovery kinetics of the current WiChR variant mean that it is appropriate for longer-term inhibition–experiments that require silencing over minutes or hours. As the first-line tool for most inhibitory optogenetics experiments in *Drosophila*, we recommend KCR1-C29D, due to its potent inhibitory effect, utility in high-chloride cells, minimal toxicity, improved potassium selectivity, and prompt recovery interval. For maximal effect, ATR should be added to food at ≥1 mM, and a light intensity ≥40 μW/mm² used. As potassium conductances are expected to hyperpolarize cells and their compartments regardless of intracellular chloride concentration, using a potassium-selective channelrhodopsin disambiguates the interpretation of the result if the chloride state is unknown. In some cases, ACR1 could be tested in conjunction with KCR1 to address questions about chloride status in a cell type. For *C. elegans* and zebrafish, we recommend that KCR2 and KCR1, respectively, be used for inhibition in cases

where intracellular chloride is unknown (or it is known that chloride levels are high). The availability of KCRs opens up the possibility to intervene in the signaling functions of hyperpolarizing high-chloride non-neuronal cells, for example, silencing excitable endocrine cells, or investigating the effects of membrane potential on signaling in non-excitable cells.

# Methods

## *Drosophila* husbandry and all-*trans*-retinal food

Flies were raised on standard cornmeal-based food containing 1.25% *w/v* agar, 10.5% *w/v* dextrose, 10.5% *w/v* maize, and 2.1% *w/v* yeast[87] at ambient temperature (24 °C). Adult flies intended for optogenetic experiments were reared in the dark and placed on food that contained all-*trans*-retinal (ATR, R2500, Sigma-Aldrich) for 2–3 days prior to experiments, as previously described[13]. Where larvae were used, (*elav-Gal4, c240-Gal4, hodor-Gal4,* and electrophysiology experiments), the parents were placed directly on ATR food. For ATR food, a stock solution of ATR was prepared in 100% ethanol (*w/v*) in the dark and mixed with warm, liquefied food to a final standard concentration (e.g., 1 mM, see concentration series). Each vial was covered with aluminum foil and stored in the dark at room temperature.

## *Drosophila* stocks

The following stocks were obtained from the Bloomington *Drosophila* Stock Center (BDSC): *Burs-Gal4* (BDSC #40972)[71], 20x-UAS-CsChrimson (BDSC #55134)[57], *OK371-Gal4* (BDSC #26160)[55]. *elav-Gal4* (BDSC #458)[88], *AstA-Gal4* (BDSC#51979)[70], and *NKCC-Gal4* (BDSC#77815)[89]. The *Gr64f-Gal4* (BDSC #57669) stock[90] was initially obtained from the BDSC and crossed out to remove balancer chromosomes and markers before being used in experiments. The *UAS-ACR1* stock was generated previously[13]. The *c240-Gal4* stock[75] was provided by Dr Daniel Cox (Georgia State University, USA). The *hodor-Gal4* stock[78] was provided by Dr. Irene Miguel-Aliaga (Imperial College London, UK). The *MB247-Gal4* stock[47,91] was provided by Dr Hiromu Tanimoto (Tohoku University, Japan).

## Fly constructs and genetics

*UAS-KCR1-ET*, *UAS-KCR2-ET*, *UAS-KCR1-GS*, and *UAS-WiChR* transgenic lines were generated by de novo synthesis (Genscript) of *Drosophila* codon-optimized HcKCR insert sequences[43] (Genbank #MZ826861 and #MZ826862) or the WiChR sequence[45] (Genbank #OP710241) as eYFP fusions. After Sanger sequencing verification (Genscript), the fragments were cloned into a *pJFRC7-20XUAS-IVS-mCD8::GFP* vector (*Addgene* plasmid #26220), replacing the *mCD8::GFP* insert via restriction enzyme digest (XhoI, XbaI). For *UAS-KCR1-GS*, a 3× GGGGS sequence was used to link the opsin with the fluorophore. For the KCR-ET and WiChR constructs, an AAA linker sequence was used as the starting point, to which two modifications were made: (1) an FCYENEV motif was added to the C terminus of eYFP to boost protein export from the endoplasmic reticulum and prevent potential aggregate formation[51]; and (2) a KSRITSEGEYIPLDQIDINV trafficking signal from Kir 2.1[52] was added to the linker at C terminus of the opsin to boost protein expression[10]. The KCR1-C29D variant[45] was obtained by site-directed mutagenesis of the KCR1-ET sequence, where the cysteine at position 29 was replaced by aspartic acid (Genscript). The synthesized constructs were injected into flies and targeted to attP1 or attP2 insertion sites on the second or third chromosomes, respectively, and the transgenic progeny were balanced either over CyO or TM6C (BestGene). Expression was verified by imaging of eYFP fluorescence with a Leica TCS SP8 STED confocal microscope. Opsin transgenic flies were crossed with relevant Gal4 driver lines to produce F1 offspring for use as test subjects. Driver Gal4 lines and UAS-opsin responder lines were each crossed with an otherwise wild-type *w[1118]* line and the F1 progeny (e.g., *UAS-KCR1-ET/+* or *elav-Gal4/+*) were used as control subjects.

## C. elegans husbandry and ATR media

C. elegans were cultured at 18°C on Nematode Growth Media (NGM) plates (1.9% w/v Bacto Agar, 0.25% w/v Bacto Peptone, and 0.3% w/v NaCl, supplemented with KPO$_4$, MgSO$_4$, CaCl$_2$ and cholesterol) seeded with a lawn of Escherichia coli strain OP50. mCherry-positive worms were selected for worm tracking assays. An ATR stock solution was prepared in 100% ethanol (w/v) in the dark and mixed with E. coli strain OP50 to a final concentration of 1 mM. This mixture was seeded onto NGM plates, wrapped in aluminum foil, and stored in the dark at room temperature.

## C. elegans constructs

gBlocks (IDT) containing codon-optimized cDNAs for the respective opsin were fused with an eYFP fluorophore at the C terminus and three synthetic introns to enhance expression. cDNAs were PCR amplified and ligated in the KpnI and EcoRI sites of an sdf-9P::mCherry vector using the following primer sets: ACR1-f and ACR-1-YFP-r for sdf-9p::ACR1::YFP; KCR-1-f and KCR-1-YFP-r for sdf-9p::KCR1::YFP; KCR-1-GS-f and KCR-1-GS-YFP-r for sdf-9p::KCR1(GS)::YFP; KCR-1-f and KCR-1-YFP-r for sdf-9p::KCR2::YFP. Genomic DNA corresponding to the pan-neuronal promoter (snt-1p) was PCR-amplified from C. elegans genomic DNA using the following primer sets: snt-1P-FseI-F and snt-1P-AscI-R, and then ligated in the FseI and AscI sites, to generate snt-1p::ACR1::YFP, snt-1p::KCR1::YFP, snt-1p::KCR1(GS)::YFP, and snt-1p::KCR2::YFP. To establish transgenic strains, the plasmid was co-injected with elt-2::mCherry at 10 ng/µl each into the gonads of adult N2 hermaphrodites using a microinjector (InjectMan 4). Complete primer, cDNA sequences, and C. elegans genotypes are given in Supplementary information (see Source Data file).

## C. elegans confocal imaging

L4 hermaphrodite worms were transferred to a glass slide and immobilized on 3% agarose pads using 2–3 µl of 1 mg/µl levamisole diluted in M9 buffer. Images were then captured under a 100× objective. Multiple transgenic lines of each transgene were examined for fluorescent protein expression and localization patterns. Spinning disc confocal microscopy was performed on a setup built around a Nikon Ti2 inverted microscope equipped with a Yokogawa CSU-W1 confocal spinning head, a Plan-Apo objective (100× 1.45 NA), and a back-illuminated sCMOS camera (Prime 95B; Photometrics). Excitation light for YFP was provided by 488 nm/150 mW (Coherent) (power measured at optical fiber end) through DPSS laser combiner (iLAS system; Gataca systems). All image acquisition and processing steps were controlled by MetaMorph (Molecular Device) software. Images were acquired with exposure times in the 400–500 ms range.

## C. elegans locomotion tracking

The worms were cultured on E. coli OP50 supplemented with 1 mM ATR for 1 day before testing. Individual worms were placed in 5 × 5 mm arenas cut into a 50 mm ∅ transparent acrylic disk planted in a 60 mm Petri dish filled with NGM (Fig. 5B). Locomotor behavior was recorded at 30 FPS surrounded by infrared lighting (850 nm) in the Spinnaker SDK application. The recording was performed with a FLIR Grasshopper3 near-infrared video camera (Edmund Optics, GS3-U3-41C6NIR-C) equipped with an 850 nm longpass filter (Green.L, 58-850) and a white diffuser in between the infrared lighting and the worms, to prevent light reflection into the camera. The arena was illuminated from the side with green (λ 530 nm, 75 µW/mm$^2$) or blue (λ 460 nm, 65 µW/mm$^2$) LEDs. The worms were allowed to roam the arena for 10 s in the dark before being exposed to 10 s of green light illumination. After illumination, each C. elegans was tracked for an additional 40 s in the dark to assess the paralysis recovery rate. Videos were down-sampled and the worms were tracked using DeepLabCut (DLC) pose-estimation neural network software[92]. The frame size was 512 × 512 pixels; the worm width and length were ~17 and ~80 pixels,

respectively. Ten key points were visually labeled along the length of the worm in 280 frames each from 14 videos to create a ground-truth dataset[93]. Of the labeled ground-truth frames, 95% were used to train a Resnet 50-based neural network over 500,000 iterations. For points with prediction confidence above a 0.6 cutoff, the root-mean-square error between ground-truth locations and predicted locations was 1.47 pixels for training and 3.97 pixels for testing, roughly 5% the length of a worm. This trained model was then used to analyze other similarly acquired videos. The mean worm speed was calculated from the raw X- and Y-coordinates of the centroid keypoints (n = 4) using a custom script written in Python. To normalize against jitter, the average speed of control animals for the first 10 s of the experiment was set to 0. This average speed value was then subtracted from each respective tracking experiment and the corrected speeds were plotted against time.

## Opsin expression in N2a cell culture

After sequence verification, ACR and KCR construct variants were cloned into the multiple cloning sites of a pcDNA3.4 vector (Genscript) by XhoI and EcoRV restriction enzyme digest. Then, 250 ng of the respective DNA constructs were transfected into N2a cells[54] using Lipofectamine 3000 (Invitrogen). The cells were left to incubate in serum-free media for 48 h. The N2a cultures were then washed three times with PBS and fixed for 20 min at room temperature with 4% paraformaldehyde diluted in PBS-Triton X-100 (0.25%, 85111 Thermo Fisher Scientific). After fixation, the cells were blocked in 5% BSA (A-420-500, Gold Biotechnology) diluted in PBS-Triton X-100 (0.25%) for 1 h at room temperature and stained for GFP (Abcam ab13970, RRID: AB_300798) at 1:2000 v/v dilution for 1.5 h at 37°C. Afterward, the cultures were rinsed three times with PBS and incubated with an Alexa 488 goat anti-chicken (A-11039 Thermo Fisher Scientific, RRID: AB_2534096) at 1:500 v/v dilution for 1 h at 37°C. Finally, the cells were washed three times with PBS and mounted onto microscope slides in Vectashield Vibrance mounting media (H-1700 Vector Laboratories, Burlingame, CA). Imaging was performed on a Zeiss LSM700 upright microscope using a 100× objective. Maximum intensity projections were obtained after image analysis with ImageJ.

## Zebrafish experiments

Confocal imaging, optogenetic illumination, and locomotion tracking in zebrafish larvae were done as previously described[15]. Briefly, the methods were as follows. For imaging experiments, 24-h-old F1 embryos were dechorionated, anesthetized with 160 mg/l (w/v) tricaine, and mounted in 1% (w/v) low melting agarose in E3. Imaging was done with a Zeiss LSM800 confocal microscope with a 10× and a 40× water immersion objective. For movement analysis, the embryos (embryos of both sexes were used) were screened with a fluorescence stereomicroscope to identify opsin-expressing fish. The chorions containing the embryos were then placed in a glass dish with 24 concave wells on a stereomicroscope (Zeiss Stemi 2000) with a transmitted light base. Behavior was recorded on the microscope using a Point Gray Flea2 camera controlled by MicroManager, as previously described[15]. Each embryo was tested once for each condition. Image analysis was carried out using Fiji (RRID:SCR_002285) and Python scripts. From each recording, one frame was extracted per second to obtain a total of 46 frames (including the first and last frames). Circular regions of interest were manually drawn around each chorion to isolate each fish. Subsequently, each frame was subtracted from the next frame to identify the differences between frames. The number of different pixels in each region of interest was taken as a measure of the movement of each embryo. Embryos that did not move during the entire recording were excluded from the analysis.

## Drosophila immunohistochemistry and confocal imaging

Primary antibodies used include mouse anti-Brp[94] (nc82, DSHB, RRID AB_2314866) at 1:50 v/v dilution, anti-Dlg[95] (4F3, DSHB,

RRIDAB_528203) at 1:50 $v/v$ dilution, and chicken anti-GFP (Ab13970 Abcam, RRID: AB_300798) at 1:2000 v/v dilution. Secondary antibodies used include Alexa 488 goat anti-chicken (A-11039 Thermo Fisher Scientific, RRID: AB_2534096) at 1:1000 $v/v$ dilution and Alexa 647 donkey anti-mouse (715-605-151, Jackson ImmunoResearch, RRID: AB_2340863) at 1:500 dilution. Adult brains were dissected in cold PBS (0.1 mM PB) and fixed with 4% paraformaldehyde for 30 min as previously described[13]. Briefly, fixed brains were washed three times in PBST (0.2% Triton-100, 85111 Thermo Fisher Scientific) and incubated in primary antibodies in PBST at 4 °C for 48 h, after which they were rinsed and incubated in secondary antibodies in PBST at 4 °C. Finally, the brains were washed three times in PBST for 15 min each and mounted on microscope slides in Vectashield Vibrance (H-1700 Vector Laboratories, Burlingame, CA) and covered with a coverslip. Slides containing mounted fly brains were viewed under a Leica TCS SP8 STED 3X or a Zeiss LSM700 upright microscope using a 20× objective. Maximum intensity projections were calculated using Leica Application Suite X software on the $z$-axis.

## Summary of antibodies
The following antibodies were used in this study:

Mouse anti-cockroach allatostatin (Ast7) (DSHB 5F10, 1:2 $v/v$ dilution)

Mouse anti-fly BRP (DSHB nc82, RRID AB_23148662, 1:50 $v/v$ dilution)

Mouse anti-fly Dlg (4F3, DSHB, RRIDAB_528203, 1:50 $v/v$ dilution)

Chicken anti-GFP (Abcam ab13970, RRID AB_300798, 1:2000 $v/v$ dilution)

Alexa 488 goat anti-chicken (A-11039 Thermo Fisher Scientific, RRID AB_2534096, 1:1000 $v/v$ dilution)

Alexa 647 donkey anti-mouse (715-605-151, Jackson ImmunoResearch, RRID AB_2340863, 1:500 $v/v$ dilution).

Alexa 568 goat anti-mouse (A-11004 Thermo Fisher Scientific, RRID: AB_2534072, 1:500 $v/v$ dilution)

The validation of the four primary antibodies is as follows. The anti-allatostatin antibody (Ast7) (DSHB 5F10, 1:2 $v/v$ dilution) was originally validated by immunohistochemistry[96] and has since been documented in 12 publications. Mouse anti-fly disks large (DSHB 4F3) were raised against the second PDZ domain of Dlg[95], and have been widely used in Western blot, immunohistochemistry, and other applications in 95 publications. Chicken anti-GFP (Abcam ab13970, RRID AB_300798, 1:2000 $v/v$ dilution) has been validated by Abcam via Western blot and immunohistochemistry. Mouse anti-fly BRP (DSHB nc82, RRID AB_23148662, 1:50 $v/v$ dilution) was originally identified to bind to the Bruchpilot protein[94] and has been documented in over 1000 publications.

## *Drosophila* AstA cells actuation and confocal imaging
To probe for cytotoxic effects induced through opsin expression, we expressed *UAS-KCR1-ET*, *UAS-CD8-GFP*, and *UAS-ACR1* with *AstA-Gal4*. *AstA-Gal4* drives expression in four cells of the *Drosophila* subesophageal zone that are positive for the neuropeptide allatostatin A[70]. After being raised on standard food, the flies were transferred to food with 0.5 mM ATR for 2–3 days and subsequently exposed to green light (31 µW/mm²) for 6 days. Afterwards, fly brains were dissected in a modified HL3 solution (described in Electrophysiology) and fixed with 4% paraformaldehyde for 30 min. Fixed brains were washed three times in PBST and incubated in 10% goat serum in PBST overnight. Brains were then incubated in 200 ul of primary antibody solution ((DSHB Cat# 5F10, RRID: AB_528076, at 1:2 $v/v$ dilution)[96,97] at 4 °C for 48 h, after which they were rinsed and incubated in 200 ul of secondary antibody solution (A-11004 Thermo Fisher Scientific, RRID: AB_2534072) at 1:500 $v/v$ dilution in PBST at 4 °C for 24 h. Brains were

visualized under a confocal microscope on an LSM710 Carl Zeiss and the number of AstA-positive cells was counted manually.

## *Drosophila* electrophysiology
Larval-nerve electrophysiological experiments were performed and analyzed as described previously[13,98]. In brief, third-instar larvae were dissected in a modified HL3 solution comprising: 110 mM NaCl, 5 mM KCl, 5 mM HEPES, 10 mM NaHCO₃, 5 mM trehalose, 30 mM sucrose, 1.5 mM CaCl₂, and 4 mM MgCl₂. An abdominal nerve was drawn into a glass electrode with a fire-polished tip[98]. Extracellular recordings from the nerve were performed with a Multiclamp 700B (Molecular Devices) and digitized with a Digidata 1440A (Molecular Devices). Data was acquired at a sampling rate of 10 kHz. Light actuation (40 µW/mm²) was induced with 0.5 s and 30 s pulses of green LED light triggered by pCLAMP 10 software (Molecular Devices). Data were excluded from the analysis if there was an absence of spiking prior to light onset. The recordings were bandpass filtered at 100–1.5 kHz before performing spike detection. To detect spikes, a window discriminator was used as previously described[13,99]. Briefly, spikes were defined as upward signals that peaked within 0.05 s and crossed the amplitude threshold, which was defined as 2.58 SD above the mean amplitude. To calculate the spike frequency, a rolling window of 100 ms and 500 ms, for 500 ms and 30 s light pulses, respectively, was used.

## *Drosophila* larvae tracking
Larvae tracking was performed in a 84 × 90 mm cassette containing 30 behavioral arenas arranged in two rows. Each arena was CNC milled with 26 × 4 mm discorectangle geometry from 1.5-mm-thick transparent acrylic and backed with a black sheet. Arenas were coated with 3% agarose. Individual third-instar larvae were loaded into each arena in the dark. Arenas were then covered with a transparent acrylic lid. The cassette was placed horizontally and illuminated with green or blue light from a mini-projector (Optoma ML750) positioned above the cassette. Behaviors were recorded under infrared (IR) light at 24 FPS. Each video frame was processed in real time and connected to CRITTA tracking software[100]. ACR1, KCR1-ET, and KCR1-C29D larvae were illuminated with green light (92 µW/mm²). WiChR larvae were illuminated with blue light (27 µW/mm²). In each experiment, the larvae were tracked in the dark for 5 min, followed by illumination for 1 min and then dark for another 5 min.

## Climbing assay
Fly climbing performance was monitored in a 170 × 94 mm acrylic cassette. The cassette contained a total of 17 individual rectangular chambers, with each chamber being 7 mm long, 86 mm high and 3 mm wide (Fig. 2A). Following ice anesthesia, one individual fly was transferred into each chamber and the chambers were closed with a transparent acrylic sheet that was sandwiched into the cassette. After the transfer, the flies were given 5 min to recover from anesthesia before the start of the experiment. Climbing behavior was recorded at 5 FPS under infrared backlighting (850 nm). The recording was performed with a Chameleon3 near-infrared video camera (FLIR CM3-U3-13S2C) equipped with a 4.4–11 mm FL High-Resolution Varifocal Lens (Edmund Optics) and an 850 nm long pass filter (Green.L, 58-850). The cassette was illuminated from the front with LEDs. During the test session, the flies were allowed to freely explore the arena. Each video frame was processed in real time and connected to CRITTA tracking software[100]. The cassette was first manually tapped downwards and then the flies were allowed to climb for 20 s in the dark (IR light only). The flies were then agitated again and allowed to climb in the dark for 3 s, following which the optogenetic light was switched on for 20 s (green at 11 µW/mm² or blue light at 85 µW/mm²). After light exposure, the flies were recorded in the dark for an additional 20 s before the experiment was concluded.

## Olfactory memory

Aversive olfactory conditioning was performed in a multifly olfactory trainer (MOT) as previously described[13,101]. Briefly, up to six flies per chamber were conditioned in a behavioral arena that was 50 mm long, 5 mm wide, and 1.3 mm high (Fig. 3E). Odors were conveyed into each end of the chambers by carrier air, as adjusted by mass-flow controllers (Sensirion AG)[102]. During 60 s of shock-odor training, the flies received 12 electric shocks at 60 V, which were delivered through the circuit boards on the floor and ceiling of the chamber. Fly behavior was recorded at 25 FPS with an AVT F-080 guppy camera that was connected to a video acquisition board (PCI-1409, National Instruments). In each experiment, the flies were conditioned to either avoid 3-octanol (OCT) or 4-methylcyclohexanol (MCH). Post-training odor preference was tested by exposing one half of the chamber to the punished odor and the other half to the unpunished odor. A performance index (PI) was calculated[103] by counting flies in individual video frames over the final 30 s of the assessment. Flies expressing a channelrhodopsin with *MB247-Gal4* were initially tested for shocked-odor avoidance in the presence of an inhibitory green light (λ 530 nm). Subsequently, to ensure that the unactuated flies had normal learning capacity, the performance of the same flies was retested in the absence of optogenetic inhibition. Avoidance in the presence of light was then compared between the control and test flies.

## Capillary feeding assay

Adult flies (5–10-days-old) were starved for 24 h at 25°C on 2% agarose in water and then tested in an automated capillary-based feeding assay[60,61,104]. Flies were briefly anesthetized on ice and loaded into acrylic feeding chambers equipped with capillaries containing liquid food (5% sucrose, 5% yeast extract, 0.5% food dye in water; see schematic in Fig. 3B). The feeding chambers were then placed between an 850 nm infrared light source and a custom infrared-filtered image-acquisition system for feeding tracking. The meniscus of the food was marked using an infrared-absorbing dye[61]. Images of the experiments were acquired using custom Vision Acquisition Software for Labview (National Instruments). Feeds were detected when a fly was detected in the feeding alcove and the food level dropped simultaneously. Evaporation of the food was tracked and subtracted from feed levels. Green light from LEDs (24 μW/mm²) was administered in two of the four epochs (On–Off–On–Off) for 30 min each. The flies were allowed to feed for 2 h and their total feed volume was quantified using custom analysis software written in Python.

## Trumelan activity-monitoring assay

The Trumelan cassette consisted of an array of chambers in two rows of 13 chambers (each 32L × 3H × 3W mm, Fig. 2I). The fly chambers were cut into a 3-mm-thick acrylic plate, backed with a second sheet, and covered in front of the chamber array with a third sheet of acrylic. A piece of matte black card was placed behind the cassette to ensure contrast. The cassette was placed vertically in a Sanyo MIR-154 incubator set to 25°C. The flies were recorded at 10 FPS with a FLIR Grasshopper3 near-infrared video camera (Edmund Optics, GS3-U3-41C6NIR-C) equipped with a 50 mm fixed-focus lens (Edmund Optics, VS-C5024-10M) and an 850 nm long pass filter (Green.L, 58-850). Two sets of infrared LED boards (850 nm peak emission) illuminated the cassette continuously. Each video frame was processed in real time with CRITTA LabView software, which calculated behavioral metrics for each fly. Experiments consisted of three epochs: 60 s of darkness (i.e., infrared illumination only), followed by 60 s LED illumination, and a final 60 s in the dark. For the green light-titration experiments, three light conditions were used: 12 μW/mm², 24 μW/mm², and 45 μW/mm². For WiChR experiments blue light illumination (24 μW/mm²) was used.

## Wing-expansion assay

Crosses were prepared on food that contained 0.5 mM ATR and then maintained in the dark. At 24 h post puparium formation, the pupal cases were transferred into a new vial and exposed to green light (31 μW/mm²) at 24 °C. Wing expansion was scored under a dissection microscope 2–3 days post eclosion.

## Eclosion assay

Fifteen virgin females (1–3-days-old) were crossed with four, 1–3-day-old males in a vial that contained standard food medium and the flies were allowed to mate for 24 h at ambient temperature (24 °C). The vial was wrapped in aluminum foil to prevent light exposure. The flies were subsequently transferred to a new vial and the procedure was repeated three more times. The number of eggs in each vial after 24 h of mating was tabulated. After all offspring eclosed, the number of flies was counted and the percentage of eclosed flies relative to the number of eggs in the vial was calculated.

## Hodor-cell optogenetics

Larval feeding chambers were coated with a thin layer of agar containing 5% sucrose and 0.5% blue food dye. Third-instar *Drosophila* larvae were allowed to feed for 30 min on the food under a lid held by magnets, under green light (38 μW/mm²). At the end of the feeding session, the larvae were briefly washed in PBS and anesthetized in 100% ethanol, before being imaged using an iPhone 12 mini equipped with a macro lens. Images were inverted and quantified in Fiji[105], where single larvae were outlined and food intake was estimated via the mean intensity of all pixels in the red channel.

## Nociception experiments

Assessment of the larvae nociceptive rolling escape response was performed as previously described[76,98,106]. Briefly, third-instar larvae expressing a channelrhodopsin with *c240-Gal4* were placed in a drop of water (30 μl) on a Petri dish. The Petri dish was placed on a hotplate that was set to 70 °C, with or without illumination by green light (λ 530 nm, 51 μW/mm²). The change in water temperature over time was measured with a thermocouple placed in the droplet next to the larva. The time for the larvae to first display the corkscrew behavior[76,98,107] was observed visually.

## Optogenetic illumination

Green (peak emission 530 nm, Luxeon Rebel, SP-05-G4, Quadica Developments Inc.) and blue (peak emission 460 nm, Luxeon Rebel, SP-05-G4, Quadica Developments Inc.) LEDs mounted on a heatsink (Luxeon Star N50-25B, Quadica Developments Inc.) were used in all optogenetic experiments. LEDs were powered and modulated by the output voltage of a 700-mA BuckPuck driver (Luxeon Star 3023-D-E-700, Quadica Developments Inc.). Unless otherwise indicated, the illumination and behavioral apparatus were placed inside a temperature-controlled incubator (MIR-154, Sanyo) throughout the experiment. Light-intensity measurements were performed as previously described[13]. Briefly, a photodiode (Thorlabs S130C) connected to a power and energy-meter console (Thorlabs PM100D) was used to measure light intensity in a dark room. The meter was zeroed before each measurement.

## Statistical analyses

Experiments were not performed in a blinded fashion. No initial power calculations were made to determine the sample sizes; significance tests were not conducted[108]. Estimation statistics were used to analyze quantitative data with the DABEST software library[109]. Mean-difference effect sizes were computed between the control and the test intervention. Bootstrap methods were applied to calculate the distributions and 95% CIs of the differences between the groups tested[109]. Data

analysis was performed and visualized with Jupyter Python notebooks calling the DABEST, pandas, scikits-bootstrap, seaborn, and SciPy packages.

## Reporting summary

Further information on research design is available in the Nature Portfolio Reporting Summary linked to this article.

## Data availability

Source data are provided with this paper. The raw data in support of the findings of this study are available from a Zenodo repository under the following link: https://doi.org/10.5281/zenodo.10648742. Source data are provided with this paper.

## Code availability

Data analysis code in support of the findings presented in this study is deposited at the Zenodo repository under the following link: https://doi.org/10.5281/zenodo.10648742.

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

## Acknowledgements

We are grateful to Dr. Tong-Wey Koh (Temasek Life Sciences Laboratory) for assistance with electrophysiology experiments. We thank James Stewart, Joses Ho, Wang Zhiping, and Deepak Choudhury for assistance with the Espresso rig. We thank Dr. Daniel Cox (Georgia State University) for providing the *subdued-Gal4* stock and Dr. Irene Miguel-Aliaga (Imperial College London) for providing the *hodor-Gal4* stock. We also thank Dr Jun Nishiyama (Duke-NUS Medical School) for providing N2a mouse neuro-blastoma cells. We acknowledge the Advanced BioImaging facility of Singhealth/DukeNUS for assistance with the generation of ima-ging data. S.O., I.H., X.Y.Z., N.L., Z.Z., and A.C.-C. were supported by grants MOE2019-T2-1-133, FY2022-MOET1-0001, and T2EP30222-0016 from the Ministry of Education, Singapore; S.O. and D.S. were supported by NMRC Young Investigator Research Grant MOH-OFYIRG20nov-0024; SX was supported by the A*STAR Scientific Scholars Fund and NMRC Young Investigator Research Grant MOH-OFYIRG20nov-0051; J.A. was supported by the A*STAR Research Attachment Program of the A*STAR Graduate Academy; N.L. was supported by Research Scholarship MOE2019-T2-1-133; R.H. and Y.S. were supported by MOE-T2EP30120-0002 from the Ministry of Education, Singapore; V.C. and S.J. were supported by MOE-T2EP30121-0017 from the Ministry of Education, Singapore; the Duke-NUS and A*STAR authors were supported by a Biomedical Research Council block grant to the Institute of Molecular and Cell Biology, and a Duke-NUS Medical School block grant to A.C.-C.

## Author contributions

Conceptualization: S.O., A.C.-C., S.X.; experiment design: S.O., A.C.-C., S.X., N.L., S.J.; methodology: S.O., S.X., and N.L.; software: S.X., D.S. (DLC implementation); data analysis: S.O., S.X., I.H., N.L., J.A., S.J.; investigation: S.O. (construct design, memory experiments, falling experiments, nociception, and eclosion experiments), S.X. (brain dis-section, immunohistochemistry and microscopy, adult feeding, and hodor experiments), I.H. (electrophysiology), N.L. (*C. elegans* tracking), D.S. (*C. elegans* ground truth annotation and tracking, N2a culture and microscopy), J.A. (horizontal activity tracking), X.Y.Z. (AstA toxicity, brain dissection, immunohistochemistry, and microscopy), W.Y. (wing expansion), Z.Z. (falling experiments, larvae tracking), R.H. (*C. elegans* constructs design and microscopy), V.C. (zebrafish constructs, beha-vior), S.J. (zebrafish imaging, behavior); writing (original draft): S.O.; writing (revision): S.O., A.C.C., S.X.; visualization: S.O., S.X., A.C.-C., J.A., N.L., S.J., I.H.; supervision: S.O., Y.S., S.J., and A.C.-C.; project adminis-tration: A.C.-C.; funding acquisition: A.C.-C.

## Competing interests

The authors declare no competing interests.

## Ethics approval

The experiments here were carried out in accordance with guidelines approved by the Institutional Animal Care and Use Committee of Bio-polis, Singapore.
