## [Peer Review File · Nature Communications]

Kalium channelrhodopsins effectively inhibit neuronsREVIEWER COMMENTS

Reviewer #1 (Remarks to the Author):

The authors characterize and apply recently identified Kalium channelrhodopsins (KCRs) in small animal model organisms including *Drosophila*, *C. elegans*, and zebrafish larvae. The current limitations of available optogenetic silencing tools, which are mainly based on anion pumps or channelrhodopsins (ACRs), have been well documented over the last few years. These include low light sensitivity and long-term toxicity (pumps), as well as depolarization instead of shunting/hyperpolarization (ACRs) in compartments/cells with inverted chloride potentials. Thus, the need for novel and well characterized silencing tools is still imminent, and in particular light activated potassium channels are well suited to overcome the current limitations. The authors drive a well conducted study that focuses on the characterization of the silencing properties of KCRs in *Drosophila*. They first optimized KCR trafficking in *Drosophila* neurons using different linkers and ER/Golgi export signals and then extensively compared their KCRs with the most commonly used ACR (gtACR1). They performed a full panel of behavioral and physiological assays in *Drosophila* (locomotion, feeding, memory, nociception), as well as proof of principle experiments in *C.elegans* and zebrafish. Importantly, they point out that KCRs allow silencing/blocking of cellular function that cannot be inhibited by ACRs due to differences in chloride concentrations. This study thus nicely demonstrates the usefulness of KCRs as an inhibitory tool in small model organisms, which will be extremely beneficial for studies addressing circuit function and/or behavior in these systems. The study is overall excellent and should be published, but I have some comments regarding the specificity of the lines/experiments and the properties of KCRs, which should be addressed:

1. The authors should clearly mention that KCRs are not fully potassium selective but also conduct sodium, which might account for some of the observations by the authors (10.1126/sciadv.add7729).
2. Along these lines, the authors mention the recent discovery of another potassium channelrhodopsin family from *Wobblia lunata* (WiCHR), which presumably has even more favorable properties regarding K⁺ selectivity and conductance (10.1126/sciadv.add7729). Did the authors test WiChR in comparison to KCRs, which might overcome the incomplete silencing phenotypes potentially arising due to partial Na⁺ conductance of KCRs (e.g. in walking assays)?
3. The representative larval nerve recordings to me look a bit abnormal: only the elav-Gal4 control shows the expected rhythmic pattern generated by coordinated motor neuron firing (e.g. 10.1523/JNEUROSCI.4749-05.2006). Most other conditions display low rhythmicity and, in some cases, elevated spike frequencies before light mediated silencing (e.g. elav-Gal4>ACR1 or KCR1-GS). Do the corresponding animals display abnormal locomotion (larvae or adult), which might be expected due to changes in motor neuron activity during development?
4. It is a bit surprising that the authors see background expression and activity in all lines except with KCR1-ET, as all the transgenes are inserted in the same genomic locus, using the same vector backbone (20xUAS), the only difference being the transgene itself. For KCR1-ET and -GS this difference is very minor so I would not expect differences in background expression and activity in this case. The authors' argument that background off-target expression is not observed for KCR1-ET thus seems a bit odd (though I do not doubt their results showing that KCR1-ET does not have strong background effects).
5. For the *C. elegans* experiment, which strain was used in the locomotion assays? Typically, lite-1 deficient animals are used due to the strong innate response to blue light. This is not evident from the description, but based on the data the animals do not respond to light illumination per se. This should be clarified as appropriate (Methods or experimental details).
6. In Fig. 7, the representative larvae look very different in size/stage and meal consumption in some

controls (ACR1/+, KCR-GS/+) seems low compared to the other controls (as also seen in the quantitative data). Is this effect due to background activity in these lines? A constitutive effect by ACR1 might prevent an acute effect due to light-mediated inhibition of enterocytes.

Reviewer #2 (Remarks to the Author):

The currently most used neuron inhibitors in model animals are anion channelrhodopsins (ACRs), natural light-gated chloride channels widely used as potent molecular tools to suppress neuron firing when illuminated. Recently, light-gated potassium-selective cation channels named kalium channelrhodopsins (KCRs) have been discovered. Previous to this submission, the first, found in *Hyphochytrium catenoides*, HcKCR1 and HcKCR2, show promise as alternative optogenetic neuron inhibitors in early measurements in rodents.

The authors of this submission expand the testing of HcKCR1 and HcKCR2 neural suppression to *Drosophila*, *C. elegans*, and zebrafish, and directly compare their potency to that of the popular neural suppressor GtACR1 proven over the past 8 years by many investigators to be a potent and versatile inhibitor of neuronal activity in *Drosophila*, zebrafish, rodents, ferret, *C. elegans*, primates, and cardiomyocytes. Their results show that the HcKCRs, although they appeared generally slightly less potent than GtACR1, are overall similarly effective at spiking suppression. They also show that the HcKCRs can be used effectively in neurons with high chloride intracellular concentrations which preclude the use of ACRs. Optogenetic studies typically target ACRs (usually GtACR1 or GtACR2) expression to neural soma-dendritic compartments to avoid possible high chloride concentrations in axons that may cause neuron photoactivation.

This manuscript is concise and well written and is timely in providing useful information on KCR properties as new optogenetics neuron suppressers. I have just two suggested changes to improve the manuscript:

1. Page 3 – Top paragraph. Following the sentence “Actuation of these HcKCRs opens in mouse brain slices⁴³.” add a sentence referring to an article from Karl Deisseroth’s and Adam Cohen’s lab in which HcKCR1 was successfully used for optogenetic neural suppression to study synaptic plasticity in brains of living mice.

The reference is: Fan LZ, et al. All-optical physiology resolves a synaptic basis for behavioral timescale plasticity. *Cell*. 2023;186:543-59. DOI: 10.1016/j.cell.2022.12.035

2. Page 11 “To investigate this discrepancy...”. “discrepancy” implies a lack of compatibility of the data, i.e. an inexplicable difference. Therefore, I suggest changing this phrase to: “To investigate this performance difference ...”

Response to Reviewers

NCOMMS-23-35402A

Reviewer #1

1. “The authors characterize and apply recently identified Kalium channelrhodopsins (KCRs) in small animal model organisms including *Drosophila*, *C. elegans*, and zebrafish larvae. The current limitations of available optogenetic silencing tools, which are mainly based on anion pumps or channelrhodopsins (ACRs), have been well documented over the last few years. These include low light sensitivity and long-term toxicity (pumps), as well as depolarization instead of shunting/hyperpolarization (ACRs) in compartments/cells with inverted chloride potentials. Thus, the need for novel and well characterized silencing tools is still imminent, and in particular light activated potassium channels are well suited to overcome the current limitations. The authors drive a well conducted study that focuses on the characterization of the silencing properties of KCRs in *Drosophila*. They first optimized KCR trafficking in *Drosophila* neurons using different linkers and ER/Golgi export signals and then extensively compared their KCRs with the most commonly used ACR (gtACR1). They performed a full panel of behavioral and physiological assays in *Drosophila* (locomotion, feeding, memory, nociception), as well as proof of principle experiments in *C.elegans* and zebrafish. Importantly, they point out that KCRs allow silencing/blocking of cellular function that cannot be inhibited by ACRs due to differences in chloride concentrations. This study thus nicely demonstrates the usefulness of KCRs as an inhibitory tool in small model organisms, which will be extremely beneficial for studies addressing circuit function and/or behavior in these systems. The study is overall excellent and should be published, but I have some comments regarding the specificity of the lines/experiments and the properties of KCRs, which should be addressed:”

> We thank the Reviewer for their kind assessment.

2. “The authors should clearly mention that KCRs are not fully potassium selective but also conduct sodium, which might account for some of the observations by the authors (10.1126/sciadv.add7729).”

> We thank the reviewer for pointing this out. We have now clearly mentioned and referenced the observation that HcKCRs also display residual sodium conductance (section titled “KCRs with improved K⁺ selectivity localize to neuronal plasma membranes”). We also addressed this by generating and validating flies with the KCR1-C29D opsin, which has previously been shown to have improved potassium selectivity¹.

3. “Along these lines, the authors mention the recent discovery of another potassium channelrhodopsin family from *Wobblia lunata* (WiChR), which presumably has even more favorable properties regarding K⁺ selectivity and conductance (10.1126/sciadv.add7729). Did the authors test WiChR in comparison to KCRs, which might overcome the incomplete silencing phenotypes potentially arising due to partial Na⁺ conductance of KCRs (e.g. in walking assays)?”

> We agree with the reviewer that the newly discovered WiChR KCR shows great potential for optogenetic applications. We have now added a new Figure 8 to the manuscript where we

examined the inhibitory efficacy of WiChR in *Drosophila*. WiChR is very effective at silencing a variety of cell populations, However, we observed that the flies displayed long recovery kinetics which limits the applicability of WiChR for certain applications. We therefore propose the KCR1-C29D variant as currently the most versatile potassium-selective optogenetic silencer for rapid application, while WiChR is suitable for minute- or hour-scale inhibition.

4. “The representative larval nerve recordings to me look a bit abnormal: only the elav-Gal4 control shows the expected rhythmic pattern generated by coordinated motor neuron firing, (e.g. 10.1523/JNEUROSCI.4749-05.2006) ². Most other conditions display low rhythmicity and, in some cases, elevated spike frequencies before light mediated silencing (e.g. elav>ACR1 or KCR1-GS). Do the corresponding animals display abnormal locomotion (larvae or adults), which might be expected due to changes in motor neuron activity during development?”

> We thank the reviewer for pointing this out. Indeed, published nerve recordings do typically show rhythmic patterns ^{2,3}. However, locomotion in both opsin-expressing adults and larvae was normal: we observed normal locomotion patterns in larvae (Figure S5B) and adults (Figure 8I1), with or without actuation. A thorough investigation of the arrhythmic firing pattern is beyond the scope of this study.

5. “It is a bit surprising that the authors see background expression and activity in all lines except with KCR1-ET, as all the transgenes are inserted in the same genomic locus, using the same vector backbone (20xUAS), the only difference being the transgene itself. For KCR1-ET and -GS this difference is very minor so I would not expect differences in background expression and activity in this case. The authors’ argument that background off-target expression is not observed for KCR1-ET thus seems a bit odd (though I do not doubt their results showing that KCR1-ET does not have strong background effects).”

> Leak expression from UAS constructs is a fairly widely acknowledged phenomenon, making the ACR1 and KCR1-GS cases not unexpected. We state “Why we see this effect with the UAS-ACR1 and UAS-KCR1-GS transgenes, but not the UAS-KCR1-ET, is unclear and requires further investigation going forward.”

6. “For the *C. elegans* experiment, which strain was used in the locomotion assays? Typically, lite-1 deficient animals are used due to the strong innate response to blue light. This is not evident from the description, but based on the data the animals do not respond to light illumination per se. This should be clarified as appropriate (Methods or experimental details).”

> We used N2 as a background strain for all the *C. elegans* experiments: “To establish transgenic strains, the plasmid was co-injected with *elt-2::mCherry* at 10ng/μl each into the gonads of adult N2 hermaphrodites using a microinjector (InjectMan 4)”. We did not use the lite-1 deficient mutant background to evaluate the properties of KCRs in our assays, and do not view it as necessary. While *C. elegans* responds very strongly to UV, it reacts to moderate-intensity blue light only very weakly ⁴. The response to UV is indeed mediated via LITE-1 photoreceptor ⁵. The LEDs used to activate KCR channels emit most of their light around 460 nm. With the intensity used

for the assay (65 $\mu\text{W}/\text{mm}^2$), we observed no major response in the locomotory behaviors of worms in control animals.

7. “In Fig. 7, the representative larvae look very different in size/stage and meal consumption in some controls (*ACR1/+*, *KCR-GS/+*) seems low compared to the other controls (as also seen in the quantitative data). Is this effect due to background activity in these lines? A constitutive effect by *ACR1* might prevent an acute effect due to light-mediated inhibition of enterocytes.”

> We thank the reviewer for this note, and agree that *KCR1-GS/+* larvae look slightly smaller than larvae from the other genotypes. Indeed, all *KCR1-GS/+* larvae looked somewhat smaller than the others. The reviewer is also correct in pointing out that meal consumption was lower in *KCR1-GS/+* and *ACR1/+* responder controls which we believe is due to the leak expression effects mentioned earlier. This observation is now specifically mentioned in the main text (Paragraph starting “**Actuating Hodor enterocytes with *KCR1-ET* reduces larval feeding**”). We can not rule out a constitutive effect mediated by *ACR1* that could reduce the baseline consumption of food in these animals. Nevertheless, *ACR/+* larvae still consumed some amount of food and we did not observe any additional knock-on reductions in *Hodor>ACR* larvae, suggesting that *ACR* expression in *Hodor* cells did not further reduce food consumption. This stands in stark contrast to *Hodor>KCR1-ET* larvae, which consumed much less food than either *KCR1-ET/+* or *ACR1/+* controls (data not shown). Therefore, we believe our conclusion remains valid.

Reviewer #2

8. The currently most used neuron inhibitors in model animals are anion channelrhodopsins (*ACRs*), natural light-gated chloride channels widely used as potent molecular tools to suppress neuron firing when illuminated. Recently, light-gated potassium-selective cation channels named kalium channelrhodopsins (*KCRs*) have been discovered. Previous to this submission, the first, found in *Hyphochytrium catenoides*, *HcKCR1* and *HcKCR2*, show promise as alternative optogenetic neuron inhibitors in early measurements in rodents.

The authors of this submission expand the testing of *HcKCR1* and *HcKCR2* neural suppression to *Drosophila*, *C. elegans*, and zebrafish, and directly compare their potency to that of the popular neural suppressor *GtACR1* proven over the past 8 years by many investigators to be a potent and versatile inhibitor of neuronal activity in *Drosophila*, zebrafish, rodents, ferret, *C. elegans*, primates, and cardiomyocytes. Their results show that the *HcKCRs*, although they appeared generally slightly less potent than *GtACR1*, are overall similarly effective at spiking suppression. They also show that the *HcKCRs* can be used effectively in neurons with high chloride intracellular concentrations which preclude the use of *ACRs*. Optogenetic studies typically target *ACRs* (usually *GtACR1* or *GtACR2*) expression to neural soma-dendritic compartments to avoid possible high chloride concentrations in axons that may cause neuron photoactivation.

This manuscript is concise and well written and is timely in providing useful information on *KCR* properties as new optogenetics neuron suppressors. I have just two suggested changes to improve the manuscript:

> We thank the Reviewer for their kind assessment.

9. “Page 3 – Top paragraph. Following the sentence “Actuation of these HcKCRs opens in mouse brain slices⁴³.” add a sentence referring to an article from Karl Deisseroth and Adam Cohen's labs in which HcKCR1 was successfully used for optogenetic neural suppression to study synaptic plasticity in brains of living mice. The reference is: Fan LZ, et al. All-optical physiology resolves a synaptic basis for behavioral timescale plasticity. *Cell*. 2023;186:543-59. DOI: 10.1016/j.cell.2022.12.035

> We thank the reviewer for highlighting this study, we have now mentioned and cited this work.

10. Page 11 “To investigate this discrepancy...”. “discrepancy” implies a lack of compatibility of the data, i.e. an inexplicable difference. Therefore, I suggest changing this phrase to: “To investigate this performance difference ...”

>We thank the reviewer for pointing this out and agree that the suggested wording is more fitting. The relevant text passage has been amended.

References

1. Vierock, J. *et al.* WiChR, a highly potassium-selective channelrhodopsin for low-light one- and two-photon inhibition of excitable cells. *Sci Adv* **8**, eadd7729 (2022).
2. Fox, L. E., Soll, D. R. & Wu, C.-F. Coordination and modulation of locomotion pattern generators in *Drosophila* larvae: effects of altered biogenic amine levels by the tyramine beta hydroxylase mutation. *J. Neurosci.* **26**, 1486–1498 (2006).
3. Mohammad, F. *et al.* Optogenetic inhibition of behavior with anion channelrhodopsins. *Nat. Methods* (2017) doi:10.1038/nmeth.4148.
4. Ward, A., Liu, J., Feng, Z. & Xu, X. Z. S. Light-sensitive neurons and channels mediate phototaxis in *C. elegans*. *Nat. Neurosci.* **11**, 916–922 (2008).
5. Gong, J. *et al.* The *C. elegans* Taste Receptor Homolog LITE-1 Is a Photoreceptor. *Cell* **168**, 325 (2017).

REVIEWERS' COMMENTS

Reviewer #1 (Remarks to the Author):

In their revised version, the authors commendably addressed all points and added substantial experiments characterizing and comparing two additional tools (KCR1-C29D and WiChR). This work will stand as the prime reference for KCRs in small model organisms, giving scientists access to these amazing new reagents and adding extensively to the optogenetic toolbox. I have no further comments except to congratulate the authors for their impressive work.

Reviewer #1 (Remarks on code availability):

code could not be accessed using the given doi. However, I am not an expert in coding so a review of the code from my side would not be very helpful.

Reviewer #2 (Remarks to the Author):

I had in the first review considered it acceptable for publication with some suggested improvement. The authors have improved the revised version according to my recommendations and I consider it ready for publication in Nature Communications. Furthermore, I congratulate the authors on an exciting, informative, and much-needed comparison of optogenetic neural inhibition by the recently discovered KCRs and their quantitative comparisons to the currently much used ACRs.

John Spudich